# Pseudo-Generalized Dynamic View Synthesis from a Video

**Xiaoming Zhao**[1,2]* **Alex Colburn**[1] **Fangchang Ma**[1] **Miguel Ángel Bautista**[1]
**Joshua M. Susskind**[1] **Alexander G. Schwing**[1]
[1]Apple     [2] University of Illinois Urbana-Champaign
`xz23@illinois.edu`; `{alexcolburn,fangchang,mbautistamartin,`
`jsusskind,ag_schwing}@apple.com`

## Abstract

Rendering scenes observed in a monocular video from novel viewpoints is a challenging problem. For static scenes the community has studied both scene-specific optimization techniques, which optimize on every test scene, and generalized techniques, which only run a deep net forward pass on a test scene. In contrast, for dynamic scenes, scene-specific optimization techniques exist, but, to our best knowledge, there is currently no generalized method for dynamic novel view synthesis from a given monocular video. To explore whether generalized dynamic novel view synthesis from monocular videos is possible today, we establish an analysis framework based on existing techniques and work toward the generalized approach. We find a pseudo-generalized process without scene-specific *appearance* optimization is possible, but geometrically and temporally consistent depth estimates are needed. Despite no scene-specific appearance optimization, the pseudo-generalized approach improves upon some scene-specific methods. For more information see project page at https://xiaoming-zhao.github.io/projects/pgdvs.

## 1 Introduction

Rendering (static or dynamic) 3D scenes from novel viewpoints is a challenging problem with many applications across augmented reality, virtual reality, and robotics. Due to this wide applicability and the compelling results of recent efforts, this area has received a significant amount of attention. Recent efforts target either static or dynamic scenes and can be categorized into scene-specific or scene-agnostic, *i.e.*, generalized, methods.[1] Scene-specific methods for static scenes, like neural radiance fields (NeRFs) (Mildenhall et al., 2020) have been continuously revised and latest advances attain compelling results. More recently, generalized methods for static scenes have also been developed and are increasingly competitive, despite the fact that they are not optimized on an individual scene (Yu et al., 2020; Wang et al., 2021; Suhail et al., 2022; Varma et al., 2023).

While this transition from scene-specific optimization to generalized methods has been in progress for static scenes, little work has studied whether generalized, *i.e.*, scene-agnostic, methods are possible for dynamic scenes. As dynamic novel view synthesis from monocular videos is a highly ill-posed task, many scene-specific optimization methods (Li et al., 2021; Xian et al., 2021; Gao et al., 2021; Li et al., 2023) reduce the ambiguities of a plethora of legitimate solutions and generate plausible ones by constraining the problem with data priors, *e.g.*, depth and optical flow (Gao et al., 2021; Li et al., 2021). With such data priors readily available and because of the significant recent progress in those fields, we cannot help but ask a natural question:

*"Is generalized dynamic view synthesis from monocular videos possible today?"*

In this work, we study this question in a constructive manner. We establish an analysis framework, based on which we work toward a generalized approach. This helps understand to what extent we can reduce the scene-specific optimization with current state-of-the-art data priors. We find

---

*Work done as part of an internship at Apple.

[1]"Generalized" is defined as no need of optimization / fitting / training / fine-tuning on *test* scenes.

Figure 1: (a) We find that it is possible to get rid of scene-specific appearance optimization for novel view synthesis from monocular videos, which could take up to hundreds of GPU hours per video, while still outperforming several scene-specific approaches on the NVIDIA Dynamic Scenes (Yoon et al., 2020). (b) Qualitative results demonstrate the feasibility of generalized approaches as our rendering quality is on-par or better (see details of the dragon balloon), even though we do not use any scene-specific appearance optimization. The face is masked out to protect privacy.

> *"A pseudo-generalized approach, i.e., no scene-specific appearance optimization, is possible, but geometrically and temporally consistent depth estimates are needed.[2]"*

Concretely, our framework is inspired from scene-specific optimization (Li et al., 2021; Xian et al., 2021; Gao et al., 2021; Li et al., 2023) and extends their underlying principles. We first render static and dynamic content separately and blend them subsequently. For the static part, we utilize a pre-trained generalizable NeRF transformer (GNT) (Varma et al., 2023). As it was originally developed for static scenes, we extend it to handle dynamic scenes by injecting masks, which indicate dynamic content, into its transformers. For the dynamic part, we aggregate dynamic content with the help of two commonly-used data priors, *i.e.*, depth and temporal priors (optical flow and tracking), and study their effects.

Applying our analysis on the challenging NVIDIA Dynamic Scenes data (Yoon et al., 2020) we find: 1) we are unable to achieve a *completely* generalized method with current state-of-the-art monocular depth estimation or tracking; but 2) with the help of *consistent* depth (CD) estimates, we can get rid of scene-specific *appearance* optimization, which could take up to hundreds of GPU hours per video, while still outperforming several scene-specific approaches on LPIPS (Zhang et al., 2018) as shown in Fig. 1. Such CD estimates can come from affordable scene-specific optimization approaches (Luo et al., 2020; Kopf et al., 2020; Zhang et al., 2021b; 2022), taking around three GPU hours per video. All in all, we are able to achieve *pseudo-generalized* dynamic novel view synthesis. We use the word '*pseudo*' due to the required scene-specific CD optimization, and '*generalized*' because of no need for costly scene-specific appearance fitting. Note, such a pseudo-generalized approach does not 'cheat' by exploiting additional information, because CD estimates are also required by many scene-specific approaches (Xian et al., 2021; Li et al., 2023). We hence refer to CD as a *sufficient* condition for generalized dynamic novel view synthesis from monocular videos.

Moreover, we verify that our finding holds when consistent depth from physical sensors is available, *e.g.*, from an iPhone LiDAR sensor, as we still outperform several scene-specific baselines with respect to mLPIPS on the DyCheck iPhone data (Gao et al., 2022a). Note, sensor depth obviates the need for scene-specific optimizations, enabling a completely generalized process.

To summarize, our contributions are:

1. We carefully study the use of current state-of-the-art monocular depth estimation and tracking approaches for generalized dynamic novel view synthesis from monocular videos, suggesting an assessment of the downstream usability for those fields.

2. We find a sufficient condition and propose a pseudo-generalized approach for dynamic novel view synthesis from monocular videos without any scene-specific appearance optimization. We verify that this approach produces higher-quality rendering than several scene-specific methods that require up to hundreds of GPU hours of appearance fitting.

## 2 RELATED WORK

**Scene-specific static novel-view synthesis.** Early work on static scene-specific novel-view synthesis reconstructs a light field (Levoy & Hanrahan, 1996; Gortler et al., 1996) or a layered representation (Shade et al., 1998). Recent works (Suhail et al., 2021; Attal et al., 2021; Wizadwongsa

---

[2]We borrow the definition of "geometrically and temporally consistent depth estimates" from Zhang et al. (2021b): "the depth and scene flow at corresponding points should be consistent over frames, and the scene flow should change smoothly in time".

et al., 2021) extend such representations to be data-driven. Also recently, neural radiance fields (NeRFs) (Mildenhall et al., 2020) were proposed to represent the scene as an implicit neural network. Follow-up works further improve and demonstrate compelling rendering results (Barron et al., 2021; 2022; Verbin et al., 2022; Barron et al., 2023). All these works focus on static scenes and require scene-specific fitting, *i.e.*, a learned NeRF cannot be generalized to unseen scenes. Different from these efforts, we focus on dynamic scenes and aim for a scene-agnostic approach.

**Generalized static novel-view synthesis.** Prior works for scene-agnostic novel-view synthesis of static scenes utilized explicit layered representations, *e.g.*, 3D photography (Shih et al., 2020) and multiplane images (Tucker & Snavely, 2020; Han et al., 2022). Works like free view synthesis (Riegler & Koltun, 2020) and stable view synthesis (Riegler & Koltun, 2021) require a 3D geometric scaffold, *e.g.*, a 3D mesh for the scene to be rendered. Recently, static scene-agnostic novel-view synthesis has received a lot of attention in the community. Prior works tackle it with either geometry-based approaches (Rematas et al., 2021; Yu et al., 2020; Wang et al., 2021; Chibane et al., 2021; Trevithick & Yang, 2021; Chen et al., 2021; Johari et al., 2022; Liu et al., 2022; Wang et al., 2022a; Du et al., 2023; Lin et al., 2023; Suhail et al., 2022; Varma et al., 2023) or geometry-free ones (Rombach et al., 2021; Kulhánek et al., 2022; Sajjadi et al., 2022b;a; Venkat et al., 2023; Bautista et al., 2022; Devries et al., 2021; Watson et al., 2023). Differently, we focus on generalized dynamic view synthesis while the aforementioned approaches can only be applied to static scenes.

**Scene-specific dynamic novel-view synthesis.** Various scene representations have been developed in this field, *e.g.*, implicit (Pumarola et al., 2021; Li et al., 2021; Xian et al., 2021), explicit (Broxton et al., 2020; Fridovich-Keil et al., 2023), or hybrid (Cao & Johnson, 2023; Attal et al., 2023). Prior works in this area can be categorized into two setups based on the input data: multiview input data (Lombardi et al., 2019; Bemana et al., 2020; Bansal et al., 2020; Zhang et al., 2021a; Wang et al., 2022b; Li et al., 2022; Attal et al., 2023; Wang et al., 2023a) and monocular input data (Park et al., 2021b; Du et al., 2021; Tretschk et al., 2021; Gao et al., 2021; Park et al., 2021a; Cao & Johnson, 2023; Fridovich-Keil et al., 2023; Li et al., 2023; Liu et al., 2023; Fang et al., 2022; Yoon et al., 2020; Büsching et al., 2023).[3] All these approaches require scene-specific optimization. Different from these efforts, we focus on a scene-agnostic method using monocular input data.

**Generalized dynamic novel-view synthesis.** To our best knowledge, few prior works target novel-view synthesis of dynamic scenes from monocular videos in a scene-agnostic way. In the field of avatar modeling, some methods are developed for generalizable novel-view synthesis for animatable avatars (Wang et al., 2023b; Kwon et al., 2023; Gao et al., 2022b). However, these approaches require multiview input and strong domain geometric priors from human templates, *e.g.*, SMPL (Loper et al., 2015). Differently, we care about dynamic novel-view synthesis in general, which domain-specific approaches are unable to handle. The concurrent work of MonoNeRF (Tian et al., 2023) tackles this challenging setting via 3D point trajectory modeling with a neural ordinary differential equation (ODE) solver. Notably, MonoNeRF only reports results for a) temporal extrapolation on training scenes; or b) adaptation to new scenes after fine-tuning. Both setups require scene-specific appearance optimization on test scenes. Thus, it is unclear whether MonoNeRF is entirely generalizable and can remove scene-specific appearance optimization. In contrast, we show improvements upon some scene-specific optimization methods without scene-specific appearance optimization.

## 3 METHODOLOGY FOR STUDYING THE QUESTION

### 3.1 OVERVIEW

We assume the input is a monocular video of $N$ frames $\{(I_i, t_i, K_i, E_i)\}_{i=1}^N$, where $I_i \in [0,1]^{H_i \times W_i \times 3}$ is an RGB image of resolution $H_i \times W_i$ recorded at time $t_i \in \mathbb{R}$ with camera intrinsics $K_i \in \mathbb{R}^{3 \times 3}$ and extrinsics $E_i \in SE(3)$. Our goal is to synthesize a view $I_{\text{tgt}} \in [0,1]^{H_{\text{tgt}} \times W_{\text{tgt}} \times 3}$, given camera intrinsics $K_{\text{tgt}} \in \mathbb{R}^{3 \times 3}$, extrinsics $E_{\text{tgt}} \in SE(3)$, and a desired time $t_{\text{tgt}} \in [t_1, t_N]$.

To study to what extent we can reduce scene-specific optimization and to develop a generalized approach, we separate rendering of static and dynamic content as shown in Fig. 2. Note that this is also a common principle exploited in prior scene-specific works, either through explicitly decomposed rendering (Li et al., 2021; Gao et al., 2021; Li et al., 2023) or via implicit regularization (Xian et al.,

---

[3]Cao & Johnson (2023); Fridovich-Keil et al. (2023) operate on either multiview or monocular input.

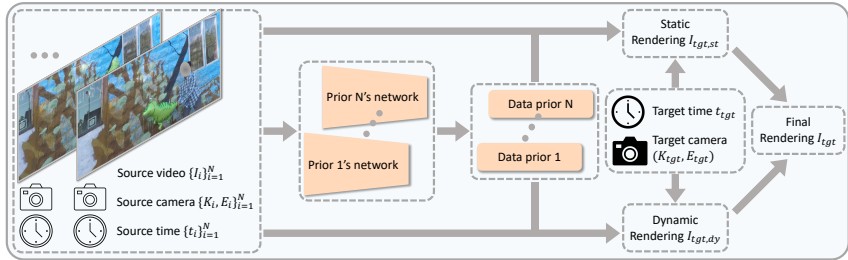

Figure 2: **Framework overview (Sec. 3.1).** Our analysis framework separately renders static (Sec. 3.2) and dynamic content (Sec. 3.3). We focus on exploiting depth and temporal data priors, which are commonly utilized in scene-specific optimization methods.

2021; Yoon et al., 2020). Formally, we obtain the target rendering as follows:

$$I_{\text{tgt}} = M_{\text{tgt, dy}} \cdot I_{\text{tgt, dy}} + (1 - M_{\text{tgt, dy}}) \cdot I_{\text{tgt, st}}, \tag{1}$$

where $M_{\text{tgt, dy}} \in \{0, 1\}^{H_{\text{tgt}} \times W_{\text{tgt}}}$ is a binary mask indicating the dynamic content on the target rendering. $I_{\text{tgt, st}}, I_{\text{tgt, dy}} \in [0, 1]^{H_{\text{tgt}} \times W_{\text{tgt}} \times 3}$ are target static and dynamic content renderings respectively.

For the static rendering, we want to benefit from the compelling progress in the field of generalized static novel view synthesis. Specifically, to handle static content in dynamic scenes, we adapt the recently proposed generalizable NeRF transformer (GNT) (Varma et al., 2023) (see Sec. 3.2 for more). For rendering of dynamic content, we study how to exploit depth and temporal data priors (see Sec. 3.3 for more). This is inspired by the common practice of scene-specific approaches which distil priors from optimization losses, *e.g.*, depth consistency (Xian et al., 2021; Li et al., 2021; 2023) and 2D-3D flow consistency (Gao et al., 2021; Li et al., 2021; 2023). Notably, to understand the limitations of data priors, we intentionally avoid any (feature-based) learning for dynamic content rendering which would hide issues caused by data priors.

## 3.2 STATIC CONTENT RENDERING

In many prior scene-specific methods for dynamic novel view synthesis from monocular videos, re-rendering the static content often already requires costly scene-specific appearance optimization. Given the recent progress of generalized static novel view synthesis, we think it is possible to avoid scene-specific appearance optimization even if the monocular video input contains dynamic objects. However, this is non-trivial. Generalized approaches for static scenes usually rely on well-defined multiview constraints, *e.g.*, epipolar geometry (Ma et al., 2004), which are not satisfied in the presence of dynamic content, *e.g.*, due to presence of new content and disappearance of existing objects. Therefore, modifications are required. For this, in this study, we carefully analyze and adapt a *pretrained* GNT (Varma et al., 2023), a transformer-based approach that is able to synthesize high-quality rendering for unseen static scenes. In the following, we briefly describe GNT's structure and our modifications. Please see Sec. A for a formal description.

GNT alternatively stacks $P = 8$ view and ray transformers to successively aggregate information from source views and from samples on a ray respectively. Concretely, to obtain the RGB values for a given pixel location in the target rendering, GNT first computes the pixel's corresponding ray $\mathbf{r}$ with the help of $K_{\text{tgt}}$ and $E_{\text{tgt}}$ (Sec. 3.1). It then uniformly samples $N_{\mathbf{r}}$ points $\{\mathbf{x}_1^{\mathbf{r}}, \dots, \mathbf{x}_{N_{\mathbf{r}}}^{\mathbf{r}}\}$ on the ray. Given desired camera extrinsics $E_{\text{tgt}}$, GNT selects $N_{\text{spatial}}$ source views with indices $S_{\text{spatial}} = \{s_j\}_{j=1}^{N_{\text{spatial}}}$. In the $p$-th iteration ($p \in \{1, \dots, P\}$), GNT first uses the $p$-th view transformer to update each sample's representation based on the raw feature from $S_{\text{spatial}}$ as well as the feature from the previous $(p-1)$-th iteration. It then utilizes the $p$-th ray transformer to exchange representations between $\{\mathbf{x}_1^{\mathbf{r}}, \dots, \mathbf{x}_{N_{\mathbf{r}}}^{\mathbf{r}}\}$ along the ray. The final RGB values for the ray $\mathbf{r}$ are obtained by transforming the aggregated feature after $P$ iterations.

### 3.2.1 GNT ADAPTATION

Our goal is to adapt a pretrained GNT so as to render high-quality static content appearing in a monocular video with dynamics. For this, we need to identify when GNT fails and what causes it. In order to build a direct connection between the GNT-produced rendering and source views, inspired by MVSNet (Yao et al., 2018), we compute the standard deviation between features from all source views for each sample $\mathbf{x}_i^{\mathbf{r}}$ and aggregate them along the ray. Fig. 3.(c).(i) and (c).(ii) provide

Figure 3: **Static content rendering (Sec. 3.2)**. Samples on a target ray (shown in (a)), project to epipolar lines in source views (see (b)). Vanilla GNT does not consider dynamic content, *i.e.*, no utilization of dynamic masks (green arrows) in (b), and produces artifacts as shown in (c).(i), *e.g.*, the rendering for static walls is contaminated by the foreground balloon. We find those artifacts correlate with the standard deviation of sampled features across source views as visualized in (c).(ii). Our proposal of using masked attention (Sec. 3.2.1) based on dynamic masks in GNT's view transformer improves the static content rendering as shown in (c).(iii). The face is masked to protect privacy.

the qualitative result when GNT is naively applied on a video with dynamics and the visualization of the standard deviation. A strong correlation between high standard deviations and low-quality renderings is apparent. In other words, GNT aims to find consistency across source views $S_{\text{spatial}}$. Due to dynamic content and resulting occlusions on some source views, GNT is unable to obtain consistency for some areas in the target rendering, resulting in strong artifacts. However, occlusions may not occur on all source views, meaning consistency can still be obtained on a subset of $S_{\text{spatial}}$.

Following this analysis, we propose a simple modification: use *masked attention* in GNT's view transformer for dynamic scenes. Essentially, if the sample $\mathbf{x}_i^{\mathbf{r}}$ projects into potentially dynamic content of a source view $I_{s_j}$, that source view will not be used in the view transformer's attention computation and its impact along the ray through the ray transformer will be reduced. As a result, the probability of finding consistency is higher. To identify potentially dynamic areas, we compute a semantically-segmented dynamic mask for each source view based on single-frame semantic segmentation and tracking with optical flow (see Sec. C.4). Fig. 3.(c).(iii) shows the result after our modification. We observe that we successfully render more high-quality static content than before. We use the adapted GNT to produce the static rendering $I_{\text{tgt, st}}$ which is needed in Eq. (1). In an extreme case, if $\mathbf{x}_i^{\mathbf{r}}$ is projected onto potentially dynamic content in all source views, it is impossible to render static content and we rely on dynamic rendering for the target pixel.

## 3.3 DYNAMIC CONTENT RENDERING

One of the key challenges in dynamic novel view synthesis from monocular videos (even for scene-specific methods) is arguably modeling of dynamic motion. This is a highly ill-posed problem as plenty of solutions exist that explain the recorded monocular video. Via regularization, prior works encourage consistency between 1) attributes induced from learned motions; and 2) generic data priors. Two typical such priors are depth (Xian et al., 2021; Li et al., 2021; 2023) and optical flow (Gao et al., 2021; Li et al., 2021; 2023). Notably, even scene-specific approaches heavily rely on such priors (Gao et al., 2021; Li et al., 2021; Gao et al., 2022a). This encourages us to exploit generic data priors for a generalized approach.

### 3.3.1 USING DEPTH AND TEMPORAL PRIORS

We utilize depth and optical flow for dynamic content rendering. For each frame $I_i$, we compute a depth estimate $D_i \in \mathbb{R}^{H_i \times W_i}$. Note, for scene-specific optimizations, learned motions can only be supervised at observed time $\{t_1, \ldots, t_N\}$. To produce renderings for temporal interpolation, *i.e.*, $t_{\text{tgt}} \notin \{t_1, \ldots, t_N\}$, previous works assume that motion between adjacent observed times is simply *linear* (Li et al., 2021; 2023). Inspired by this, for any desired time $t_{\text{tgt}}$, we first find two temporally closest source views $i_{\text{tgt}}^- = \max\{i | t_i \leq t_{\text{tgt}}, i \in \{1, \ldots, N\}\}$ and $i_{\text{tgt}}^+ = \min\{i | t_i \geq t_{\text{tgt}}, i \in \{1, \ldots, N\}\}$. We then compute two associated point clouds from $I_{i_{\text{tgt}}^-}$ and $I_{i_{\text{tgt}}^+}$ with the help of dynamic masks, depths, and optical flows. Formally, we use point clouds $\mathcal{P} =$

$$\left\{ \left( \text{Lift}_{i_{\text{tgt}}^-}(\mathbf{u}_1), \text{Lift}_{i_{\text{tgt}}^+}(\mathbf{u}_2) \right) \middle| M_{i_{\text{tgt}}^-, \text{dy}}[\mathbf{u}_1] > 0, M_{i_{\text{tgt}}^+, \text{dy}}[\mathbf{u}_2] > 0, \text{Flow}_{i_{\text{tgt}}^- \leftrightarrow i_{\text{tgt}}^+}(\mathbf{u}_1, \mathbf{u}_2) \right\}, \quad (2)$$

where $\mathbf{u}_1 \in [0, W_{i_{\text{tgt}}^-}] \times [0, H_{i_{\text{tgt}}^-}]$ and $\mathbf{u}_2 \in [0, W_{i_{\text{tgt}}^+}] \times [0, H_{i_{\text{tgt}}^+}]$ mark pixel coordinates. $\mathbb{R}^3 \ni \text{Lift}_i(\mathbf{u}) = D_i[\mathbf{u}] \cdot E_i^{-1} \cdot K_i^{-1} \cdot \mathbf{u}^\top$ computes a 2D coordinate $\mathbf{u}$'s corresponding 3D point. Here

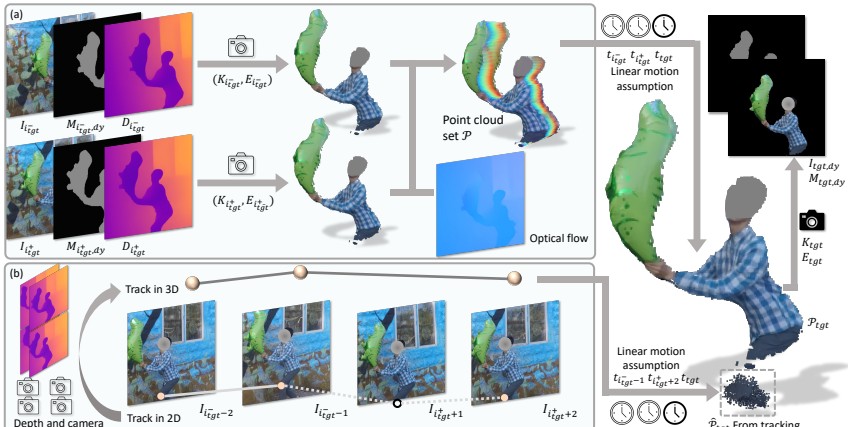

Figure 4: **Dynamic content rendering (Sec. 3.3)**. **(a)** For two temporally closest source views (indices $i_{\text{tgt}}^-$ and $i_{\text{tgt}}^+$), with the help of depth, dynamic mask, as well as camera information, 2D dynamic content, *e.g.*, human and balloon in this figure, is lifted into two point clouds. They are then connected based on optical flow between $I_{i_{\text{tgt}}^-}$ and $I_{i_{\text{tgt}}^+}$ (Sec. 3.3.1). Correspondences are highlighted as "rainbow" lines in the point cloud set $\mathcal{P}$. We then obtain the point cloud $\mathcal{P}_{\text{tgt}}$ for target time $t_{\text{tgt}}$ based on the linear motion assumption. **(b)** We utilize temporal priors of 2D tracking and lift *visible* trajectories to 3D with depth priors (Sec. 3.3.2). A complementary target point cloud is obtained based on the linear motion assumption as well and aggregated with $\mathcal{P}_{\text{tgt}}$. For example, for the shown 3D track, we linearly interpolated 3D positions from $i_{\text{tgt}}^+ - 1$ and $i_{\text{tgt}}^+ + 2$ as they are temporally closest to $i_{\text{tgt}}$ in the visible tracking (track on $i_{\text{tgt}}^+ + 1$ is invisible). The face is masked to protect privacy.

we slightly abuse notation by omitting homogeneous coordinates. $D_i[\mathbf{u}]$ denotes the interpolated depth value at coordinate $\mathbf{u}$. $M_{i,\text{dy}}[\mathbf{u}]$ is the interpolated value at $\mathbf{u}$ from the semantically-segmented dynamic mask $M_{i,\text{dy}}$ (Sec. 3.2.1), whose positive value denotes that $\mathbf{u}$ belongs to potentially dynamic content. Besides, $\texttt{Flow}_{i\leftrightarrow j}(\mathbf{u}_1, \mathbf{u}_2)$ indicates that coordinates $\mathbf{u}_1$ on $I_i$ and $\mathbf{u}_2$ on $I_j$ are connected by optical flow and they pass a cycle consistency check. Essentially, $\mathcal{P}$ refers to a paired set of two point clouds, lifting dynamic content from 2D to 3D (see Fig. 4.(a)).

For target time $t_{\text{tgt}}$, based on the linear motion assumption, we obtain dynamic content by interpolating between the two point clouds in $\mathcal{P}$ if $t_{i_{\text{tgt}}^-} \neq t_{i_{\text{tgt}}^+}$:

$$\mathcal{P}_{\text{tgt}} = \left\{ \frac{t_{\text{tgt}} - t_{i_{\text{tgt}}^-}}{t_{i_{\text{tgt}}^+} - t_{i_{\text{tgt}}^-}} \cdot \mathbf{x}_2 + \frac{t_{i_{\text{tgt}}^+} - t_{\text{tgt}}}{t_{i_{\text{tgt}}^+} - t_{i_{\text{tgt}}^-}} \cdot \mathbf{x}_1 \middle| (\mathbf{x}_1, \mathbf{x}_2) \in \mathcal{P} \right\}. \tag{3}$$

In the case of $t_{i_{\text{tgt}}^-} = t_{i_{\text{tgt}}^+} = t_{\text{tgt}}$, $\mathcal{P}$ contains two identical point clouds and we define $\mathcal{P}_{\text{tgt}} = \{\mathbf{x}_1 | (\mathbf{x}_1, \mathbf{x}_2) \in \mathcal{P}\}$. Further, we conduct statistical outlier removal for $\mathcal{P}_{\text{tgt}}$ to mitigate inaccuracies in depths and masks. We then render the point cloud $\mathcal{P}_{\text{tgt}}$ with either SoftSplat (Niklaus & Liu, 2020) or point/mesh-based renderer to obtain the dynamic RGB $I_{\text{tgt, dy}}$ and mask $M_{\text{tgt, dy}}$ in Eq. (1). See Sec. C.3 for more details about the renderer.

### 3.3.2 USING MORE TEMPORAL PRIORS

Sec. 3.3.1 uses at most two source views to render dynamic objects. To study the use of information from more views we track a dynamic object's motion and integrate tracked points for final rendering based on the linear motion assumption. Please see Fig. 4.(b) and Sec. B for more details.

## 4 EXPERIMENTS

We aim to answer: 1) how far is the performance from scene-specific methods (Sec. 4.2); 2) to what extent can scene-specific optimization be reduced by which priors (Sec. 4.3).

### 4.1 EXPERIMENTAL SETUP

**Dataset.** We conduct quantitative evaluations on the NVIDIA Dynamic Scenes data (Yoon et al., 2020) and the DyCheck iPhone data (Gao et al., 2022a). The former consists of eight dynamic

---

[4]TiNeuVox needs consistent depth to compute 3D points of static background for the background loss: https://github.com/hustvl/TiNeuVox/blob/d1f3adb/lib/load_hyper.py#L79-L83.

Table 1: **Results on NVIDIA Dynamic Scenes (13992 images).** LPIPS is reported by multiplying with 1000. Results for Row 1 - 2 are from Li et al. (2023). Results for Row 3 - 5 are reproduced by us (see Sec. E for clarifications). Row 6-1 and 6-2 evaluate on checkpoints after 20k and 40k steps respectively. DVS, NSFF, DynIBaR, TiNeuVox, and ours utilize the same consistent depth (CD) estimates.[4] Though originally DVS and NSFF only utilize monocular depth, for a fair comparison, Li et al. (2023) retrain DVS and NSFF with CD estimates instead. Appearance fitting time and hardware information are either transcribed from corresponding papers or provided by authors.

| | | Appearance Fitting (h) | Appearance Fitting Hardware | Full Image | | | Dynamic Area | | | Static Area | | |
|---|---|---|---|---|---|---|---|---|---|---|---|---|
| | | | | PSNR↑ | SSIM↑ | LPIPS↓ | PSNR↑ | SSIM↑ | LPIPS↓ | PSNR↑ | SSIM↑ | LPIPS↓ |
| 1-1 | TiNeuVox (20k) (Fang et al., 2022) | 0.75 | 1 V100 | 18.94 | 0.581 | 254.8 | 16.87 | 0.468 | 316.6 | 19.45 | 0.591 | 245.7 |
| 1-2 | TiNeuVox (40k) (Fang et al., 2022) | 1.5 | 1 V100 | 18.86 | 0.578 | 244.2 | 16.77 | 0.464 | 292.3 | 19.38 | 0.588 | 237.0 |
| 2 | Nerfies (Park et al., 2021a) | 16 | 8 V100s | 20.64 | 0.609 | 204.0 | 17.35 | 0.455 | 258.0 | - | - | - |
| 3 | HyperNeRF (Park et al., 2021b) | 8 | 4 TPUv4 | 20.90 | 0.654 | 182.0 | 17.56 | 0.446 | 242.0 | - | - | - |
| 4 | DVS (Gao et al., 2021) | 36 | 1 V100 | 27.96 | 0.912 | 81.93 | 22.59 | 0.777 | 144.7 | 29.83 | 0.930 | 72.74 |
| 5 | NSFF (Li et al., 2021) | 48 | 2 V100s | 29.35 | 0.934 | 62.11 | 23.14 | 0.784 | 158.8 | 32.06 | 0.956 | 46.73 |
| 6 | DynIBaR (Li et al., 2023) | 48 | 8 A100s | 29.08 | 0.952 | 31.20 | 24.12 | 0.823 | 62.48 | 31.68 | 0.971 | 25.81 |
| 7 | Ours | 0 | Not Needed | 26.15 | 0.922 | 64.29 | 20.64 | 0.744 | 104.4 | 28.34 | 0.947 | 57.74 |

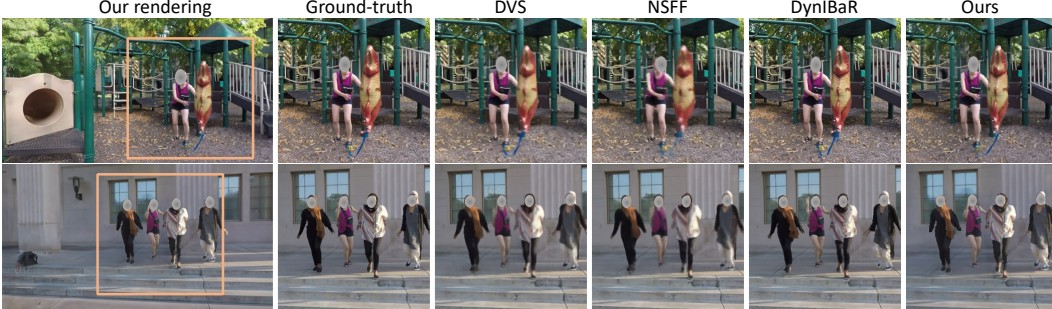

Figure 5: **Qualitative results on NVIDIA Dynamic Scenes.** Even without scene-specific appearance optimization, the method still produces on-par or higher quality renderings than some scene-specific approaches: our background is aligned well with the ground-truth, our highlighted foreground is sharper than DVS and NSFF, and we do not miss the right arm as DynIBaR does in the bottom row. Faces are masked to protect privacy.

scenes captured by a synchronized rig with 12 forward-facing cameras. Following prior work (Li et al., 2023), we derive monocular videos by selecting frames at different time steps in a round robin manner. The resulting monocular video covers most timesteps. Note, this is different from the setup in (Li et al., 2021; Gao et al., 2021) where only 24 timesteps from the full video are selected. We evaluate novel view synthesis with respect to the remaining 11 held-out views at each time step. The DyCheck iPhone data contains seven dynamic scenes captured by three synchronized cameras: one hand-held moving iPhone and two stationary cameras with a large baseline. Following Gao et al. (2022a), we evaluate on frames captured from the latter two viewpoints.

**Preprocessing.** For the NVIDIA dynamic scenes data, we use COLMAP (Schönberger & Frahm, 2016) to estimate camera poses. When consistent depth (CD) is needed, we utilize dynamic video depth estimates (Zhang et al., 2021b). For the DyCheck iPhone data, all camera poses and depths are obtained from the iPhone sensors. For both datasets, we compute optical flow with RAFT (Teed & Deng, 2020) and obtain a semantically-segmented dynamic mask with OneFormer (Jain et al., 2023) and Segment-Anything-Model (Kirillov et al., 2023) (see Sec. C.4 for details).

**Implementations.** To render static content, we directly utilize GNT's pretrained weights. The possibility to re-use pretrained weights is one advantage of our GNT adaptation (Sec. 3.2.1). We emphasize that the pretrained GNT is never exposed to the dynamic scenes we are evaluating on. Please see Sec. C for more implementation details.

**Baselines and evaluation metrics.** We compare to scene-specific methods to assess whether scene-specific optimization can be reduced. On both datasets we compare to Nerfies (Park et al., 2021a), HyperNeRF (Park et al., 2021b), NSFF (Li et al., 2021), and TiNeuVox (Fang et al., 2022). We run TiNeuVox on each video for 40k steps, double the default of 20k iterations. This gives TiNeuVox an additional advantage. On NVIDIA Dynamic Scenes data, we additionally compare to DVS (Gao et al., 2021) and DynIBaR (Li et al., 2023). We use PSNR, SSIM (Wang et al., 2004) and LPIPS (Zhang et al., 2018) on the full image, dynamic area, and static area following prior work (Li

Table 2: **Results on DyCheck iPhone data (3928 images).** T-NeRF, Nerfies, and HyperNeRF are supervised by metric depth from the iPhone LiDAR during optimization.[5] TiNeuVox needs metric depth to compute a loss on background 3D points during optimization. Appearance fitting time, hardware information, and results for row 1 - 4 are from Gao et al. (2022a). Row 5-1 and 5-2 evaluate on checkpoints after 20k and 40k steps respectively. Due to the small effective multi-view factors (EMFs) (Gao et al., 2022a), all approaches (including ours) struggle.

| | | Appearance Fitting (h) | Appearance Fitting Hardware | mPSNR↑ | mSSIM↑ | mLPIPS↓ |
|---|---|---|---|---|---|---|
| 1-1 | TiNeuVox (20k) (Fang et al., 2022) | 0.75 | 1 V100 | 14.03 | 0.502 | 0.538 |
| 1-2 | TiNeuVox (40k) (Fang et al., 2022) | 1.5 | 1 V100 | 13.94 | 0.500 | 0.532 |
| 2 | NSFF (Li et al., 2021) | 24 | 4 A4000s / 2 A100s | 15.46 | 0.551 | 0.396 |
| 3 | T-NeRF (Gao et al., 2022a) | 12 | 4 A4000s / 2 A100s | 16.96 | 0.577 | 0.379 |
| 4 | Nerfies (Park et al., 2021a) | 24 | 4 A4000s / 2 A100s | 16.45 | 0.570 | 0.339 |
| 5 | HyperNeRF (Park et al., 2021a) | 72 | 1 A4000 / 1 A100 | 16.81 | 0.569 | 0.332 |
| 6 | Ours | 0 | Not Needed | 15.88 | 0.548 | 0.340 |

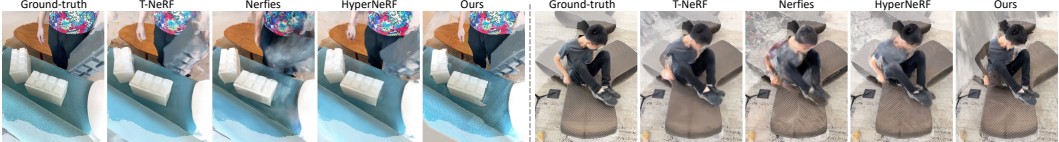

Figure 6: **Qualitative results on DyCheck iPhone data.** Due to the small effective multi-view factors (EMFs) (Gao et al., 2022a), all approaches struggle to produce high-quality renderings, which is reflected in Tab. 2. However, ours still produces on-par or better renderings than scene-specific ones: background aligns well with the ground-truth and foreground is sharper. Whitish areas indicate regions that are not considered during evaluation as they are invisible in training frames.

et al., 2023). On the DyCheck iPhone data, we also compare to T-NeRF (Gao et al., 2022a) and report masked PSNR (mPSNR), masked SSIM (mSSIM), and masked LPIPS (mLPIPS) following the dataset protocol (see Gao et al. (2022a) Appendix C). For all metrics, to avoid a result skewed by the scene with the most frames, we first compute the averaged metric over all frames for a scene, and then compute the mean across all scenes' averaged metrics.

## 4.2 COMPARISON TO SCENE-SPECIFIC APPROACHES

Tab. 1 and 2 summarize results on the NVIDIA Dynamic Scenes and DyCheck iPhone data. We compare our study-found best performing model, which utilizes consistent depth, to scene-specific methods. As expected, a generalized method does not improve upon all scene-specific baselines due to the huge disadvantage of *no* scene-specific appearance optimization compared with 100 to 384 GPU hours of fitting in scene-specific baselines. However, it is encouraging that a generalized method can perform on par or even better than some baselines on LPIPS, a major indicator of perceptual quality (Zhang et al., 2018). Fig. 5 and 6 show qualitative results. Note, compared to baselines we do not use additional information as the same consistent depth priors are used during optimization of Tab. 1's DVS, NSFF, DynIBaR, and TiNeuVox and Tab. 2's T-NeRF, Nerfies, HyperNeRF, and TiNeuVox.

## 4.3 ABLATIONS

We aim to understand what contributes to the achievement of reducing scene-specific optimizations presented in Sec. 4.2. For this we ablate on the NVIDIA dynamic scenes data (Tab. 3). Here we only list the main ablations. Please refer to Sec. D for more extensive ablation studies.

**Consistent depth is not enough.** Although our best-performing model (row 3) utilizes consistent depth (CD), CD itself is not enough for high-quality rendering. This can be concluded by comparing row 1 *vs*. 2 in Tab. 3, differing in the static rendering strategy. In row 1, we lift depths for static areas in each frame to point clouds and aggregate them for a 3D representation for static regions in the scene. Then a static image $I_{st}$ is produced via point-based rendering. In row 2, we use the vanilla GNT without any adaptation for static rendering. The main reason for inferior performance of row 1 stems from the sparsity of depth and inaccuracies for background/static regions while data-driven and ray-marching based GNT bypasses this issue.

---

[5]NSFF is not supervised with metric depth due to its NDC formulation. See Sec. 5.2 "Benchmarked results" in Gao et al. (2022a) for more details.

Table 3: **Ablations on NVIDIA Dynamic Scenes ([Yoon et al., 2020](#)) (13992 images).** LPIPS is reported by multiplying with 1000. 'Masked Input' means masking out dynamic content in input frames; 'Masked Attention' is from Sec. 3.2.1; 'CD' means consistent depth and ZoeD means depth from [Bhat et al. (2023)](#); '$\mathcal{P}_{tgt}$ Clean' marks statistical outlier removal (see Sec. 3.3.1); 'Track' indicates the method for dynamic tracking. Please see Tab. S4 for more ablations.

| | GNT | Masked Input | Masked Attention | Depth | $\mathcal{P}_{tgt}$ Clean | Track | Full Image | | | Dynamic Area | | | Static Area | | |
|---|---|---|---|---|---|---|---|---|---|---|---|---|---|---|---|
| | | | | | | | PSNR↑ | SSIM↑ | LPIPS↓ | PSNR↑ | SSIM↑ | LPIPS↓ | PSNR↑ | SSIM↑ | LPIPS↓ |
| 1 | ✗ | ✗ | ✗ | CD | ✓ | ✗ | 21.10 | 0.717 | 211.2 | 19.35 | 0.688 | 167.0 | 21.78 | 0.726 | 212.8 |
| 2 | ✓ | ✗ | ✗ | CD | ✓ | ✗ | 25.86 | 0.919 | 69.70 | 20.86 | 0.744 | 110.6 | 27.49 | 0.943 | 63.46 |
| 3 | ✓ | ✗ | ✓ | CD | ✓ | ✗ | 26.15 | 0.922 | 64.29 | 20.64 | 0.744 | 104.4 | 28.34 | 0.947 | 57.74 |
| 4 | ✓ | ✗ | ✓ | ZoeD | ✓ | ✗ | 21.15 | 0.814 | 142.3 | 15.93 | 0.479 | 233.5 | 23.36 | 0.854 | 129.9 |
| 5-1 | ✓ | ✗ | ✓ | CD | ✓ | TAPIR | 25.87 | 0.917 | 70.34 | 20.61 | 0.736 | 114.8 | 27.67 | 0.942 | 63.37 |
| 5-2 | ✓ | ✗ | ✓ | CD | ✓ | CoTracker | 25.80 | 0.917 | 69.65 | 20.51 | 0.731 | 117.6 | 27.57 | 0.942 | 62.53 |

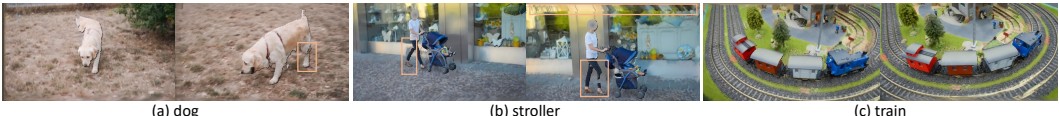

| (a) ZoeD | (b) w/ ZoeD | (c) w/ TAPIR | (d) Ours | (e) Ground-truth |

Figure 7: **Qualitative results for ablations.** (a) two views of a point cloud from aggregating 5 frames's ZoeDepth: apparent inconsistency; (b) rendering with ZoeDepth: duplicated content; (c) rendering with tracking with TAPIR: floating artifacts. Faces are masked to protect privacy.

| (a) dog | (b) stroller | (c) train |

Figure 8: **Qualitative results on DAVIS.** Our approach is able to produce reasonable renderings on various scenes. Highlighted artifacts of missing parts in foreground objects and a blurry background suggest future work on context-aware refinement. Faces are masked to protect privacy.

**GNT adaptation is needed.** This is verified by row 3's superiority over row 2. Note the difference for the static region. Please see Fig. 3.(c) for qualitative results on improved rendering in static areas.

**Consistent depth is necessary** as verified by row 3 *vs.* 4 in Tab. 3 and Fig. 7.(b). In row 4, we replace the consistent depth with predictions from ZoeDepth ([Bhat et al., 2023](#)), the state-of-the-art single-image metric depth estimator. We further mitigate depth scale-shift ambiguities by aligning 1) depth induced from the sparse point cloud from COLMAP's structure-from-motion; and 2) ZoeDepth's predictions on corresponding 2D projections of sparse point clouds. Nevertheless, we still observe inferior performance with ZoeDepth. We think the inconsistencies across frames shown in Fig. 7.(a) are the root cause, indicating room for monocular depth estimation improvements.

**Simply using tracking is not good enough** as validated by row 4 *vs.* 5-1/5-2 and Fig. 7.(c). In Row 5-1/5-2, we utilize state-of-the-art point tracking approaches TAPIR ([Doersch et al., 2023](#)) and CoTracker ([Karaev et al., 2023](#)) to aggregate dynamic content information from more frames. We notice consistent performance drops in Row 5-1 and 5-2. We hypothesize it is because 1) inaccuracies are accumulated by chaining tracking and depth estimates; and 2) our linear motion assumption is too coarse when aggregating long-term dynamic information across several frames, indicating the need for a sophisticated design to obtain accurate temporal information aggregation.

### 4.4 MORE QUALITATIVE RESULTS ON DAVIS

We render spatio-temporal interpolation videos on DAVIS ([Perazzi et al., 2016](#)), where we use COLMAP to estimate camera poses and consistent depth estimates ([Zhang et al., 2021b](#)). Qualitative results in Fig. 8 demonstrate the capabilities of our framework to handle various scenes. Meanwhile, we observe artifacts for 1) missing parts in the foreground caused by occlusions in source views or inaccurate depths; and 2) blurriness in the background stemming from out-of-boundary sampling in source views. These suggest context-aware inpaining could be a valuable future work.

## 5 CONCLUSION

We study the problem of generalized dynamic novel view synthesis from monocular videos. We find a pseudo-generalized method which avoids costly scene-specific appearance training if consistent depth estimates are used. Improvements in monocular depth estimation and tracking are needed.

## ACKNOWLEDGEMENTS

We thank Zhengqi Li for fruitful discussions and for providing rendered images from DVS and NSFF for evaluations in Tab. 1.

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

# APPENDIX – PSEUDO-GENERALIZED DYNAMIC VIEW SYNTHESIS FROM A VIDEO

This appendix is organized as follows:

## A  FORMAL DESCRIPTION OF STATIC RENDERING

Here we provide a formal description of the GNT analysis and our adaptation discussed in Sec. 3.2.

### A.1  GNT REVISITED

GNT (Varma et al., 2023) is a transformer-based approach that synthesizes a high-quality rendering for unseen static scenes. It alternatingly stacks $P = 8$ view and ray transformers to successively aggregate information from source views and from samples on a ray respectively. Formally, to obtain the corresponding RGB values for a given pixel location in the target rendering, GNT first computes its corresponding ray $\mathbf{r}$ with the help of $K_{\text{tgt}}$ and $E_{\text{tgt}}$ (Sec. 3.1). It then uniformly samples $N_{\mathbf{r}}$ points $\{\mathbf{x}_1^{\mathbf{r}}, \ldots, \mathbf{x}_{N_{\mathbf{r}}}^{\mathbf{r}}\}$ on the ray. Hereafter, we omit the superscript $\mathbf{r}$ for simplicity. A pixel's RGB value is computed as $C_{\mathbf{r}} = f_{\text{ToRGB}}\left(\frac{1}{N_{\mathbf{r}}}\sum_{i=1}^{N_{\mathbf{r}}}\mathbf{F}_{\mathbf{x}_i,P}\right)$, where $f_{\text{ToRGB}}$ is a fully-connected layer. $\mathbf{F}_{\mathbf{x}_i,P} \in \mathbb{R}^d$ marks the $d$-dimensional feature for $\mathbf{x}_i$ computed via the $P$-th ray transformer. Specifically, we have

$$\{\mathbf{F}_{\mathbf{x}_i,p}, a_{\mathbf{x}_i,p}\}_{i=1}^{N_{\mathbf{r}}} = \mathcal{T}_{\text{Ray},p}\left(\left\{\widehat{\mathbf{F}}_{\mathbf{x}_i,p}\right\}_{i=1}^{N_{\mathbf{r}}}\right), \; \forall p \in \{1, \ldots, P\}, \tag{S1}$$

where $\mathcal{T}_{\text{Ray},p}(\cdot)$ is the $p$-th ray transformer based on self-attention and $a_{\mathbf{x}_i,p}$ is the attention value, *i.e.*, $a_{\mathbf{x}_i,p} \in [0,1]$ and $\sum_{i=1}^{N_{\mathbf{r}}} a_{\mathbf{x}_i,p} = 1$ (see discussion by Varma et al. (2023) in Sec. E). $\widehat{\mathbf{F}}_{\mathbf{x}_i,p} \in \mathbb{R}^d$ is the $d$-dimensional feature from the $p$-th view transformer. Essentially, given desired camera extrinsics $E_{\text{tgt}}$ (Sec. 3.1), GNT selects $N_{\text{spatial}}$ source views with indices $S_{\text{spatial}} = \{s_j\}_{j=1}^{N_{\text{spatial}}}$. Each sample $\mathbf{x}_i$ extracts its own information $\widehat{\mathbf{F}}_{\mathbf{x}_i,p}$ from $S_{\text{spatial}}$ and the ray transformer aims to exchange such information along the ray. To obtain $\widehat{\mathbf{F}}_{\mathbf{x}_i,p}$, we have

$$\widehat{\mathbf{F}}_{\mathbf{x}_i,p} = \mathcal{T}_{\text{View},p}\left(\mathbf{F}_{\mathbf{x}_i,p-1}, \left\{\widetilde{\mathbf{F}}_{\mathbf{x}_i,s_j}\right\}_{j=1}^{N_{\text{spatial}}}\right), \; \forall p \in \{1, \ldots, P\}, \tag{S2}$$

where $\mathcal{T}_{\text{View},p}(\cdot, \cdot)$ denotes the $p$-th view transformer, and the first and second arguments are for query and key / value respectively. $\mathbb{R}^d \ni \widetilde{\mathbf{F}}_{\mathbf{x}_i,s_j} = f_{\text{RGB}}(I_{s_j})\left[\Pi(\mathbf{x}_i, E_{s_j}, K_{s_j})\right]$ is $\mathbf{x}_i$'s feature vector extracted from the $j$-th closest source view $I_{s_j}$. Here $f_{\text{RGB}}(\cdot)$ is a feature extractor and $\Pi(\mathbf{x}_i, E_{s_j}, K_{s_j})$ marks the perspective projection of $\mathbf{x}_i$ to $I_{s_j}$. $a[b]$ denotes interpolating value from $a$ based on coordinate $b$. In essence, the view transformer updates each sample's attached information based on the raw feature $\widetilde{\mathbf{F}}_{\mathbf{x}_i,s_j}$ as well as $\mathbf{F}_{\mathbf{x}_i,p-1}$ that considers communication along the ray in the previous iteration. Note, GNT initializes sample features as $\mathbb{R}^d \ni \mathbf{F}_{\mathbf{x}_i,0} = \texttt{ElemMax}(\{\widetilde{\mathbf{F}}_{\mathbf{x}_i,s_j}\}_{j=1}^{N_{\text{spatial}}})$ where $\texttt{ElemMax}(\cdot)$ refers to element-wise maximum pooling.

### A.2  GNT ADAPTATION

Our goal for adapting a pretrained GNT is to render static content appearing in a monocular video in as high of a quality as possible. For this, we need to understand when GNT fails and what causes the failure. In order to build a direct connection between the GNT-produced rendering and source views,

inspired by MVSNet (Yao et al., 2018), we compute the standard deviation between features from all source views for each sample and aggregate them along the ray using attention value from Eq. (S1):

$$\sigma_{\mathbf{r},p} = \sum_{i=1}^{N_{\mathbf{r}}} a_{\mathbf{x}_i,p} \cdot \mathtt{std} \left( \left\{ f_{\mathcal{T}_{\mathrm{View},p},\mathrm{Key}} \left( \widetilde{\mathbf{F}}_{\mathbf{x}_i,s_j} \right) \right\}_{j=1}^{N_{\mathrm{spatial}}} \right), \tag{S3}$$

where $f_{\mathcal{T}_{\mathrm{View},p},\mathrm{Key}}(\cdot)$ is the projection of the "key" in the $p$-th view transformer and $\mathtt{std}(\cdot)$ is the standard deviation computation. This essentially mimics the volume rendering equation to aggregate point-wise attributes similar to the discussion by Varma et al. (2023) in their Sec. E.

Fig. 3.(c).(i) and (ii) shows a qualitative example when naively applying GNT on a dynamic video and the visualization of $\sigma_{\mathbf{r},p}$. We can easily observe a strong correlation between high standard deviations and low-quality renderings. In other words, GNT essentially finds consistency across source views $S_{\mathrm{spatial}}$. As a result, due to the dynamic content and its resulting occlusions in some source views, GNT is unable to find consistency for some areas in the target rendering, resulting in strong artifacts. However, occlusions may not occur on all source views, which means consistency can still be obtained on a subset of $S_{\mathrm{spatial}}$. Following this analysis, we propose a simple adaptation by using *masked attention* in GNT's view transformer. Concretely, we modify Eq. (S2) to

$$\widehat{\mathbf{F}}_{\mathbf{x}_i,p} = \mathcal{T}_{\mathrm{View},p} \left( \mathbf{F}_{\mathbf{x}_i,p-1}, \left\{ \widetilde{\mathbf{F}}_{\mathbf{x}_i,j}, \mathbb{1}(\xi_{\mathbf{x}_i,s_j} > 0) \right\}_{j=1}^{N_{\mathrm{spatial}}} \right), \forall p \in \{1, \ldots, P\}, \tag{S4}$$

where $\mathbb{1}(\cdot)$ is an indicator function. $\xi_{\mathbf{x}_i,s_j} = M_{s_j,\mathrm{dy}} \left[ \Pi(\mathbf{x}_i, E_{s_j}, K_{s_j}) \right]$ is the interpolated value from a semantically-segmented dynamic mask $M_{s_j,\mathrm{dy}}$, which is computed from $I_{s_j}$ with One-Former (Jain et al., 2023) and Segment-Anything-Model (Kirillov et al., 2023) (see Sec. C.4). Essentially, $\xi_{\mathbf{x}_i,s_j} > 0$ indicates that the sample $\mathbf{x}_i$ projects into the potentially dynamic content on source view $I_{s_j}$. Consequently, $\widetilde{\mathbf{F}}_{\mathbf{x}_i,j}$ will not be used in the view transformer's attention computation and its impact on the communication along the ray through the ray transformer will be reduced. As a result, the probability of finding consistency will increase. In an extreme case, if $\xi_{\mathbf{x}_i,s_j} > 0$ for all source views, it is impossible to render static content and we rely on dynamic rendering for the target pixel. As a result, we just return to the unmodified version that does not use any masked attention for $\mathbf{x}_i$. Fig. 3.(c).(iii) showcases the result after our modification. We can see that we successfully render more static content in high quality than before. We use the adapted GNT to produce the static rendering $I_{\mathrm{tgt, st}}$ in Eq. (1).

## B    DETAILS FOR RENDERING WITH MORE TEMPORAL PRIORS

We provide a detailed description for utilizing more temporal priors as mentioned in Sec. 3.3.2. In this work, we study the effect of generalized track-any-point (TAP) approaches due to recent impressive progress (Doersch et al., 2022; 2023; Karaev et al., 2023).

Specifically, we select $N_{\mathrm{temporal}}$ temporally-close source views. Their indices make up a set $S_{\mathrm{temporal}}$. Given a pixel coordinate in one of the source views's dynamic areas, we run 2D tracking and obtain a 2D temporal tracking trajectory:

$$\tau_{\mathrm{2D}} = \{ (\mathbf{u}_j, v_j, j) \mid j \in S_{\mathrm{temporal}} \}, \tag{S5}$$

where $j$ denotes the source view's index. $\mathbf{u}_j \in [0, W_j] \times [0, H_j]$ is the 2D coordinate on source view $I_j$. $v_j \in \{0, 1\}$ is a binary value indicating whether the tracking is visible. $v_j = 0$ means the target is occluded or out-of-boundary on $I_j$, *e.g.*, Fig. 4.(b)'s tracking on $I_{i_{\mathrm{tgt}}^+ + 1}$. We lift this 2D temporal trajectory's *visible* parts to 3D with the $\mathtt{Lift}(\cdot)$ function defined in Eq. (2). However, such a 3D temporal trajectory does not contain 3D position at $t_{\mathrm{tgt}}$, which we need to estimate. Intuitively, in absence of any additional information, the temporally closer the time to $t_{\mathrm{tgt}}$, the more likely the time's corresponding 3D position is similar to that at $t_{\mathrm{tgt}}$. Following this, we find two 3D positions on the trajectory whose time is closest to $t_{\mathrm{tgt}}$ and estimate 3D positions for $t_{\mathrm{tgt}}$ based on the linear motion assumption (similar to Eq. (3)). We repeat the above procedure for all pixels in the dynamic mask areas in all source views of $S_{\mathrm{temporal}}$ and obtain a complementary point cloud $\widehat{\mathcal{P}}_{\mathrm{tgt}}$ to $\mathcal{P}_{\mathrm{tgt}}$. Combining $\widehat{\mathcal{P}}_{\mathrm{tgt}}$ and $\mathcal{P}_{\mathrm{tgt}}$, we can conduct dynamic content rendering to obtain the dynamic RGB $I_{\mathrm{tgt, dy}}$ and mask $M_{\mathrm{tgt, dy}}$ in Eq. (1).

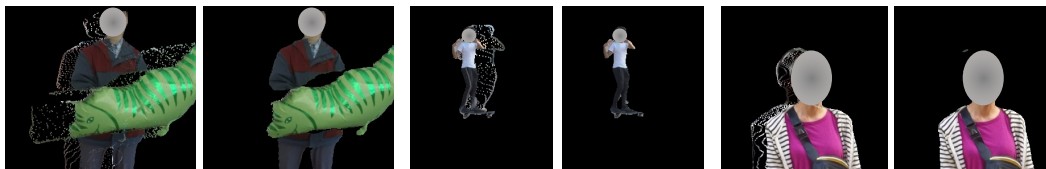

Figure S1: **Effects of statistical outlier removal.** For each scene, we show the dynamic rendering 1) without outlier removal on the left; 2) with outlier removal on the right. Apparently, statistical outlier removal produces much cleaner results. Faces are masked to protect privacy.

## C  IMPLEMENTATION DETAILS

Here we provide implementation details: we describe 1) the spatial source view selection strategy for GNT in Sec. C.1; 2) how to conduct statistical outlier removal for $\mathcal{P}_{\text{tgt}}$ in Sec. C.2; 3) the renderer for dynamic content in Sec. C.3; 4) how to compute the semantically-segmented dynamic mask in Sec. C.4; 5) how to compute consistent depth estimates in Sec. C.5; 6) how to set depth range for GNT's samples on the ray in Sec. C.6.

### C.1  SPATIAL SOURCE VIEW SELECTION FOR GNT

As mentioned in Sec. 3.2 and Sec. A, GNT selects $N_{\text{spatial}}$ source views with indices $S_{\text{spatial}} = \{s_j\}_{j=1}^{N_{\text{spatial}}}$. Here we provide the details about how to select $S_{\text{spatial}}$. Throughout our experiments, we have $N_{\text{spatial}} = 10$.

**For NVIDIA Dynamic Scenes data (Yoon et al., 2020)**, we choose the spatially closest source views. As mentioned in Sec. 4, NVIDIA Dynamic Scenes data is collected with a rig of 12 cameras and the monocular video is composed by selecting one of the 12 cameras at different time steps in a round robin manner (Li et al., 2021; 2023). Therefore, for any target camera $E_{\text{tgt}}$ with time $t_{\text{tgt}}$, we only consider source views from time $\lfloor t_{\text{tgt}} - 12 \rfloor$ to $\lceil t_{\text{tgt}} + 12 \rceil$. Without such a time window, $S_{\text{spatial}}$ will always be chosen from frames captured at the same camera. Concretely, we have $S_{\text{spatial}} = \{s_j\}_{j=1}^{N_{\text{spatial}}} = \text{NN}(E_{\text{tgt}}, \{E_i\}_{i=\lfloor t_{\text{tgt}}-12 \rfloor}^{\lceil t_{\text{tgt}}+12 \rceil}, N_{\text{spatial}})$ and $s_j \in \{\lfloor t_{\text{tgt}} - 12 \rfloor, \ldots, \lceil t_{\text{tgt}} + 12 \rceil\}, \forall j$. Here, $\text{NN}(a, b, c)$ denotes the indices of the top $c$ elements in $b$ that are spatially closest to $a$.

**For the DyCheck iPhone data (Gao et al., 2022a)**, 1) we first cluster all cameras of a monocular video into $N_{\text{cluster}}$ ($N_{\text{cluster}} \geq N_{\text{spatial}}$) groups with K-means (MacQueen, 1967) operating on positions of the cameras. 2) we detect $N_{\text{spatial}}$ clusters that are spatially closest to the target camera $E_{\text{tgt}}$ based on distances between cluster centers and the position of the target camera; 3) for each of the selected $N_{\text{spatial}}$ clusters, we choose the member that is temporally closest to $t_{\text{tgt}}$. Such a design is motivated by the small effective multi-view factors (EMFs) (Gao et al., 2022a) of the dataset: if we simply select the spatially closest source views, projected epipolar lines in all source views will capture very similar content, meaning consistency can be found almost everywhere along the epipolar lines. As discussed in Sec. 3.2.1 and shown in Fig. 3.(c), GNT's rendering relies on finding consistency across source views. However, 'Consistency almost everywhere' causes low-quality rendering. Our source view selection mitigates this situation. In this work, we use $N_{\text{cluster}} = 40$ in all our experiments.

### C.2  STATISTICAL OUTLIER REMOVAL FOR $\mathcal{P}_{\text{TGT}}$

Here we explain how to use statistical outlier removal to clean up $\mathcal{P}_{\text{tgt}}$ as mentioned in Sec. 3.3.1: 1) for each point $\mathbf{v} \in \mathcal{P}_{\text{tgt}}$, we average the distances between $\mathbf{v}$ and its $N_{\text{nn}} = 50$ nearest neighbours, *i.e.*, we obtain $N_{\text{nn}} = 50$ distances per point and compute their mean to obtain $|\mathcal{P}_{\text{tgt}}|$ averaged distances in total; 2) we compute the median $\text{median}_{\mathcal{P}_{\text{tgt}}}$ and standard deviation $\text{std}_{\mathcal{P}_{\text{tgt}}}$ of the $|\mathcal{P}_{\text{tgt}}|$ averaged distances to understand their statistics; 3) for any point $\mathbf{v} \in \mathcal{P}_{\text{tgt}}$, if its averaged distance to $N_{\text{nn}}$ nearest neighbours is larger than $\text{median}_{\mathcal{P}_{\text{tgt}}} + \delta \cdot \text{std}_{\mathcal{P}_{\text{tgt}}}$, we mark it as an outlier and remove it. Throughout our experiments, we use $\delta = 0.1$. Niklaus et al. (2019) utilizes a learned neural network to achieve similar outlier removal effects. On the contrary, we rely on point cloud's statistical properties. Fig. S1 illustrates the effect of outlier removal.

Table S1: **Ablations on rendering mechanism for dynamic content on NVIDIA Dynamic Scenes (**Yoon et al., 2020**) (13992 images).** LPIPS is reported by multiplying with 1000. Row 3 corresponds to Tab. 3's corresponding row.

| | Dynamic Renderer | Full | | | Dynamic | | | Static | | |
|---|---|---|---|---|---|---|---|---|---|---|
| | | PSNR↑ | SSIM↑ | LPIPS↓ | PSNR↑ | SSIM↑ | LPIPS↓ | PSNR↑ | SSIM↑ | LPIPS↓ |
| 3 | SoftSplat | 26.15 | 0.922 | 64.29 | 20.64 | 0.744 | 104.4 | 28.34 | 0.947 | 57.74 |
| 3-1 | PointRenderer | 25.61 | 0.916 | 65.67 | 19.97 | 0.707 | 110.6 | 27.87 | 0.944 | 58.66 |
| 3-2 | MeshRenderer | 25.78 | 0.919 | 65.63 | 19.77 | 0.709 | 111.7 | 28.32 | 0.947 | 58.40 |

Table S2: **Ablations on rendering mechanism for dynamic content on DyCheck iPhone data (**Yoon et al., 2020**) (3928 images).** LPIPS is reported by multiplying with 1000. Row 5 corresponds to Tab. 2's corresponding row.

| | Dynamic Renderer | mPSNR↑ | mSSIM↑ | mLPIPS↓ |
|---|---|---|---|---|
| 5 | PointRenderer | 15.88 | 0.548 | 0.340 |
| 5-1 | SoftSplat | 15.83 | 0.550 | 0.342 |

## C.3 RENDERER FOR DYNAMIC CONTENT

Here we describe choices of dynamic renderers. We study three types of renderers:

**Splatting-based renderer.** Using the target camera intrinsics $K_{tgt}$ and extrinsics $E_{tgt}$, we obtain the pixel coordinates for each point in $\mathcal{P}_{tgt}$ via a projection onto the target image plane. Meanwhile, since $\mathcal{P}_{tgt}$ is obtained by lifting the temporally closest source views $I_{i_{tgt}^-}$ and $I_{i_{tgt}^+}$, we also know the source view's pixel coordinate for each point in $\mathcal{P}_{tgt}$. As a result, we can connect 1) pixels in the dynamic area of the target image; and 2) pixels in the dynamic area of the source view. This enables us to use SoftSplat (Niklaus & Liu, 2020) to transfer RGB values for dynamic content from the source views to the target view. To determine the weights of splatting, we use weights of 'importance metric' type (see Eq. (14) by Niklaus & Liu (2020)). In our experiments, we set the coefficient $\alpha$ for the 'importance metric' to 100 throughout all experiments.

**Point-based renderer.** Since $\mathcal{P}_{tgt}$ is a point-based representation, it intrinsically supports point-based rendering. In our experiments, we utilize PyTorch3D's renderer with alpha-composition (Ravi et al., 2020). We set the point radius to 0.01.

**Mesh-based renderer.** Since $\mathcal{P}_{tgt}$ is computed by lifting dynamic content pixels from temporally closest source views, we can utilize the topology between pixels to convert $\mathcal{P}_{tgt}$ into a mesh. Concretely, we split every four neighboring pixels that form a rectangle into two triangles. The topology of triangles across 2D pixels can be directly transferred to $\mathcal{P}_{tgt}$. We then utilize PyTorch3D's mesh rasterizer (Ravi et al., 2020) to render the image.

Tab. S1 and Tab. S2 present ablations of renders on NVIDIA Dynamic Scenes and DyCheck iPhone data. We observe that the rendering quality doesn't differ much. We choose the best-performing one for each dataset, *i.e.*, splatting-based rendering for NVIDIA Dynamic Scenes and point-based rendering for DyCheck iPhone data.

## C.4 COMPUTATION OF SEMANTICALLY-SEGMENTED DYNAMIC MASK

Here we provide details regarding the semantically-segmented dynamic mask mentioned in Sec. 3.2.1 and Sec. 4.1.

The goal is to obtain a binary-valued dynamic mask $M_i$ for the $i$-th frame. We can use single-frame semantic segmentation to compute potential dynamic masks. Specifically, we pre-select a set of semantic labels as potentially dynamic objects, *e.g.*, person. We provide the semantic labels we use in Tab. S3. After obtaining semantic segmentations, we treat areas with pre-selected labels as dynamic and everything else as static. In this study, we run OneFormer (Jain et al., 2023) twice to cover more dynamic objects: one is trained on ADE20K (Zhou et al., 2017) while the other is trained on COCO (Lin et al., 2014). This is helpful since ADE20K and COCO have different

Table S3: **Semantic labels for potential dynamic categories**.

| Dataset | Categories |
|---------|------------|
| ADE20K | person, car, boat, bus, truck, airplane, dress/clothes, van, ship, toy, bag, motorbike, cradle, ball, animal, bicycle, fan, flag |
| COCO | person, car, motorcycle, airplane, bus, train, truck, boat, bird, cat, dog, horse, sheep, cow, elephant, bear, zebra, giraffe, umbrella, ski, snowboard, skateboard, surfboard, tennis racket |

semantic labels. We let $\widehat{M_i} \in \{0, 1\}^{H \times W}$ refer to the binary-valued mask of the $i$-th frame obtained from this procedure.

As we rely on the dynamic mask for lifting of dynamic objects to 3D, we need high-precision masks. However, if we naively set $M_i = \widehat{M_i}$, we find 1) single-frame semantic segmentation can miss dynamic objects in some frames while the same model detects the same object successfully in other frames; 2) the mask boundary precision of pure semantic segmentation does not meet our needs.

To address the first issue, we develop a streaming-based mask tracking approach to temporally propagate mask information via optical flow. We keep track of how many times a pixel is classified as being dynamic via a real-valued matrix $\widetilde{M_i}$. Specifically, 1) when $i = 0$, we have $\widetilde{M_0} = \texttt{float}(\widehat{M_0})$, where $\texttt{float}$ denotes converting the binary mask to a floating-point-based matrix. 2) when $i > 0$, $M_i = \widehat{M_i} \text{ AND } \left( \texttt{Warp}_{(i-1) \to i}(\widetilde{M}_{i-1})/(i-1) \geq \delta \right)$, where $\texttt{Warp}_{(i-1) \to i}(\cdot)$ warps mask tracking information from frame $i-1$ to frame $i$. Hence, $\texttt{Warp}_{(i-1) \to i}(\widetilde{M}_{i-1})/(i-1)$ represents a normalized value in $[0, 1]$. A value of 1 indicates that this pixel is always treated as containing dynamic content in previous frames while a value of 0 denotes that the pixel always belonged to static regions. We then use a threshold $\delta$ to convert the floating-point-based matrix to a binary matrix. Such information is useful because taking the 'AND' operation with $\widehat{M_i}$ permits to correct mistakes made by single-frame semantic segmentation. The smaller $\delta$, the higher the chance that the current $\widehat{M_i}$ will be corrected by the tracking information. In all experiments, we set $\delta = 0.5$. Finally, we update the tracking with $\widetilde{M_i} = \widetilde{M}_{i-1} + \texttt{float}(M_i)$.

To address the second issue, we combine $M_i$ computed above with segmentations predicted by the Segment-Anything-Model (SAM) (Kirillov et al., 2023). Since SAM does not predict semantic labels, we rely on $M_i$ to relate SAM's predicted segmentations to semantic categories. Specifically, we create a new binary-valued mask $M_{\text{SAM},i}$. For each of the segmentations predicted by SAM we assess its overlap with dynamic areas in $M_i$. For each segmentation from SAM, if more than 10% of its area overlaps with dynamic areas in $M_i$, we use a value of one in $M_{\text{SAM},i}$ to highlight this area. We find using $M_{\text{SAM},i}$ leads to more precise boundaries. As we use $M_{\text{SAM},i}$ as the final dynamic mask, we update the tracking information with $\widetilde{M_i} = \widetilde{M}_{i-1} + \texttt{float}(M_{\text{SAM},i})$ instead of $\widetilde{M_i} = \widetilde{M}_{i-1} + \texttt{float}(M_i)$ as stated in previous paragraph.

## C.5    CONSISTENT DEPTH ESTIMATION

Here we describe how to obtain consistent depth for monocular videos. We use work by Zhang et al. (2021b) to complete this processing, but modify two steps: 1) we replace MiDaS (Ranftl et al., 2022) with DPT (Ranftl et al., 2021) for better quality; 2) we don't change camera poses. Originally, as a preprocessing, Zhang et al. (2021b) modify the provided camera translation to align 1) neural-network predicted depth; and 2) depth induced from sparse structure-from-motion (SfM) point clouds. However, we wish to keep the camera poses unchanged for convenient rendering and evaluation. Therefore, following Li et al. (2023), we align the neural network predicted depth and depth obtained from SfM by computing a scale and shift value. We always use such precomputed scale and shift values during training.

Table S4: **Ablations on NVIDIA Dynamic Scenes (Yoon et al., 2020) (13992 images).** Row numbers are aligned with Tab. 3. LPIPS is reported by multiplying with 1000. 'Masked Input' means masking out dynamic content in input frames; 'Masked Attention' is from Sec. 3.2.1; 'CD' means consistent depth and ZoeD means depth from Bhat et al. (2023); '$\mathcal{P}_{\text{tgt}}$ Clean' marks statistical outlier removal in Sec. 3.3.1; 'Track' indicates the method for dynamic tracking.

| | GNT | Masked Input | Masked Attention | Depth | $\mathcal{P}_{\text{tgt}}$ Clean | Track | Full Image | | | Dynamic Area | | | Static Area | | |
|---|---|---|---|---|---|---|---|---|---|---|---|---|---|---|---|
| | | | | | | | PSNR↑ | SSIM↑ | LPIPS↓ | PSNR↑ | SSIM↑ | LPIPS↓ | PSNR↑ | SSIM↑ | LPIPS↓ |
| 0 | ✓ | ✗ | ✗ | ✗ | ✗ | ✗ | 24.90 | 0.906 | 97.85 | 17.79 | 0.589 | 264.0 | 28.38 | 0.947 | 74.53 |
| 1-1 | ✗ | ✗ | ✗ | CD | ✗ | ✗ | 21.08 | 0.713 | 220.2 | 19.61 | 0.674 | 192.6 | 21.68 | 0.723 | 220.2 |
| 1 | ✗ | ✗ | ✗ | CD | ✓ | ✗ | 21.10 | 0.717 | 211.2 | 19.35 | 0.688 | 167.0 | 21.78 | 0.726 | 212.8 |
| 2-1 | ✓ | ✗ | ✗ | CD | ✗ | ✗ | 25.44 | 0.909 | 82.27 | 20.66 | 0.717 | 141.6 | 26.83 | 0.934 | 74.36 |
| 2 | ✓ | ✗ | ✗ | CD | ✓ | ✗ | 25.86 | 0.919 | 69.70 | 20.86 | 0.744 | 110.6 | 27.49 | 0.943 | 63.46 |
| 3-1 | ✓ | ✓ | ✗ | CD | ✗ | ✗ | 23.75 | 0.895 | 96.62 | 19.08 | 0.679 | 182.3 | 24.99 | 0.923 | 85.72 |
| 3-2 | ✓ | ✓ | ✗ | CD | ✓ | ✗ | 23.17 | 0.899 | 86.25 | 18.00 | 0.680 | 155.8 | 24.72 | 0.927 | 77.21 |
| 3-3 | ✓ | ✓ | ✓ | CD | ✓ | ✗ | 24.93 | 0.914 | 72.41 | 19.31 | 0.717 | 126.5 | 27.10 | 0.941 | 64.60 |
| 3 | ✓ | ✗ | ✓ | CD | ✓ | ✗ | 26.15 | 0.922 | 64.29 | 20.64 | 0.744 | 104.4 | 28.34 | 0.947 | 57.74 |
| 4 | ✓ | ✗ | ✓ | ZoeD | ✓ | ✗ | 21.15 | 0.814 | 142.3 | 15.93 | 0.479 | 233.5 | 23.36 | 0.854 | 129.9 |
| 5-1 | ✓ | ✗ | ✓ | CD | ✓ | TAPIR | 25.87 | 0.917 | 70.34 | 20.61 | 0.736 | 114.8 | 27.67 | 0.942 | 63.37 |
| 5-2 | ✓ | ✗ | ✓ | CD | ✓ | CoTracker | 25.80 | 0.917 | 69.65 | 20.51 | 0.731 | 117.6 | 27.57 | 0.942 | 62.53 |

## C.6 DEPTH RANGE SELECTION FOR SAMPLES ON THE RAY

In general, we compute the depth range for the target view based on depth maps corresponding to spatial source views $S_{\text{spatial}}$ selected in Sec. C.1 with the help of camera extrinsics. Specifically, for each spatial source view's depth map, we compute a point cloud in the world coordinate system. Then we transform the point cloud into the target view's camera coordinate system. Repeating the above procedure for all spatial source views, we attain a set of depth values $D_{S_{\text{spatial}} \to \text{tgt}}$ for the target view.

**For NVIDIA Dynamic Scenes data**, we use $0.8 \times \min D_{S_{\text{spatial}} \to \text{tgt}}$ as the near value and $1.2 \times$ `Quantile`$\{D_{S_{\text{spatial}} \to \text{tgt}}, 90\}$ as the far value, where `Quantile`$(\cdot, q)$ indicates $q\%$ quantile value of the first argument.

**For the DyCheck iPhone data**, we use `Quantile`$\{D_{S_{\text{spatial}} \to \text{tgt}}, 10\}$ as the near value and `Quantile`$\{D_{S_{\text{spatial}} \to \text{tgt}}, 90\}$ as the far value. Additionally, since we have depth from the physical sensor in this dataset, we simplify the rendering by setting the depth range for the static part to be centered around its depth in the target view. To be concrete, with the help of the dynamic mask computed from Sec. C.4, we can easily identify depth values for the static part from $D_{S_{\text{spatial}} \to \text{tgt}}$ computed above. We project the transformed point cloud in the target view to the image plane to pin down the corresponding pixels for the static part. We then just set the depth range for those pixels as an interval of length $2\mathrm{e}{-4}$ centered around the depth value in the target view.

## D ADDITIONAL ABLATIONS

In Tab. S4 we provide more complete ablation studies complementary to Sec. 4.3.

**Statistical outlier removal for $\mathcal{P}_{\text{tgt}}$ is useful.** This is consistently verified by row 1 *vs*. 1-1, row 2 *vs*. 2-1, and row 3-2 *vs*. 3-1, indicating again that the computed dynamic masks and depth estimates are not yet accurate enough for the task of novel view synthesis. See Fig. S1 for qualitative results.

**Masked attention for GNT adaptation is superior.** To obtain high-quality renderings of static areas from GNT operating on videos with dynamic content, we explore three ways: 1) directly masking dynamic objects in the source views (row 3-2): this results in apparent artifacts of black boundaries as shown in Fig. S2.(a); 2) applying both masked attention and masked input (row 3-3): this still causes the black boundary artifacts but mitigates it when compared to only using masked input. See Fig. S2.(b) for more details. 3) using only masked attention (row 3): this produces the most compelling rendering.

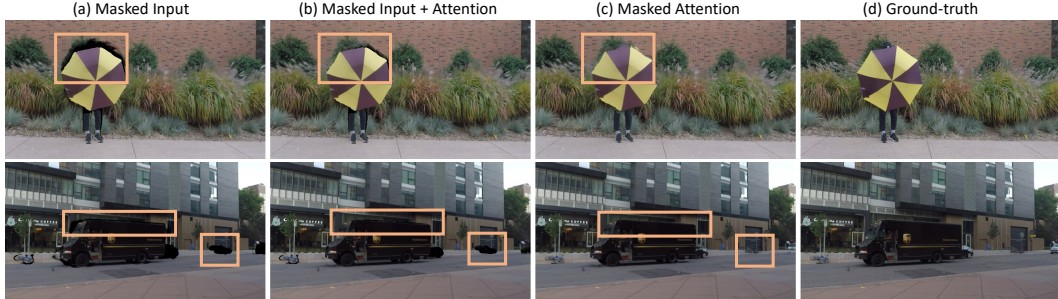

Figure S2: **Different strategies for adapting GNT.** We explore various ways to adapt GNT to dynamic scenes. (a) simply masking out dynamic content on source views results in strong artifacts like empty boundaries; (b) combining masked input and masked attention slightly alleviates the artifacts; (c) using masked attention solely produces the most compelling rendering.

Table S5: **Clarifications of baseline results on NVIDIA Dynamic Scenes (13992 images).** LPIPS is reported by multiplying with 1000. "Reproduced" means the results are obtained from running DynIBaR's official code with pretrained checkpoints or rendered images provided by authors. "DynIBaR's Mask" refers to DynIBaR's binary rendering masks, whose pixel values of true indicate that DynIBaR is able to produce content for that pixel. "?" denotes that it is unknown since it is not mentioned in the paper.

| | Approach | Type | w/ DynIBaR's Mask | Full Image | | | Dynamic Area | | | Static Area | | |
|---|---|---|---|---|---|---|---|---|---|---|---|---|
| | | | | PSNR↑ | SSIM↑ | LPIPS↓ | PSNR↑ | SSIM↑ | LPIPS↓ | PSNR↑ | SSIM↑ | LPIPS↓ |
| 1-1 | | From the paper (v1/2) | ? | 26.05 | 0.913 | 81.00 | 22.63 | 0.778 | 144.0 | - | - | - |
| 1-2 | DVS | From the paper (v3) | ? | 27.44 | 0.921 | 70.00 | 22.63 | 0.778 | 144.0 | - | - | - |
| 1-3 | | Reproduced | ✓ | 27.96 | 0.912 | 81.63 | 22.59 | 0.777 | 144.6 | 29.84 | 0.931 | 72.23 |
| 1-4 | | Reproduced | ✗ | 27.96 | 0.912 | 81.93 | 22.59 | 0.777 | 144.7 | 29.83 | 0.930 | 72.74 |
| 2-1 | | From the paper | ? | 28.90 | 0.927 | 62.00 | 23.08 | 0.783 | 159.0 | - | - | - |
| 2-2 | NSFF | Reproduced | ✓ | 29.34 | 0.934 | 61.90 | 23.15 | 0.784 | 158.7 | 32.07 | 0.956 | 46.35 |
| 2-3 | | Reproduced | ✗ | 29.35 | 0.934 | 62.11 | 23.14 | 0.784 | 158.8 | 32.06 | 0.956 | 46.73 |
| 3-1 | | From the paper | ? | 30.86 | 0.957 | 27.00 | 24.24 | 0.824 | 62.00 | - | - | - |
| 3-2 | DynIBaR | Reproduced | ✓ | 30.77 | 0.957 | 26.88 | 24.21 | 0.824 | 61.73 | 34.21 | 0.976 | 20.98 |
| 3-3 | | Reproduced | ✗ | 29.08 | 0.952 | 31.20 | 24.12 | 0.823 | 62.48 | 31.68 | 0.971 | 25.81 |

# E    CLARIFICATION FOR BASELINE RESULTS ON NVIDIA DYNAMIC SCENES

Here we clarify the difference for baseline results in Tab. 1 and the original paper (Li et al., 2023). We transcribe baseline results from Li et al. (2023). However, we find:

1) DynIBaR is unable to render full images in some cases. The official code only evaluates areas of renderings where DynIBaR produces content. It hence does not necessarily evaluate on full images.[6] We argue this favors DynIBaR, especially for "full image" and "static area" evaluation. In contrast, for "dynamic area" evaluation, the dynamic mask already eliminates the need for evaluation on areas that are not rendered by DynIBaR. Tab. S5 Row 3-2 *vs.* 3-3 verifies this hypothesis: the difference between evaluating *with or without* DynIBaR's rendering mask is more apparent on "full image" and "static area" evaluations. Note, for DVS and NSFF, the results are maintained no matter whether we use DynIBaR's mask or not (see row 1-3 *vs.* 1-4 and 2-2 *vs.* 2-3). This is because DVS and NSFF are able to render full images. Based on this finding, in Tab. 1, we use Row 3-3 from Tab. S5 for DynIBaR results since Tab. 1 conducts evaluation on full images.

2) There are discrepancies between quantitative results of *full image evaluations* for DVS from different public versions of Li et al. (2023): row 1-1 and 1-2 in Tab. S5 are for results from version 1/2 and 3 respectively.[7] We find that our reproduced results in row 1-3's "full image" evaluation section do not match any of the public versions entirely. Specifically, our reproduced PSNR (row 1-3) aligns well with the corresponding PSNR from v3 (row 1-2) while our reproduced SSIM and

---

[6]https://github.com/google/dynibar/blob/02b164/eval_nvidia.py#L383-L396
[7]v1: https://arxiv.org/pdf/2211.11082v1.pdf; v2: https://arxiv.org/pdf/2211.11082v2.pdf; v3: https://arxiv.org/pdf/2211.11082v3.pdf.

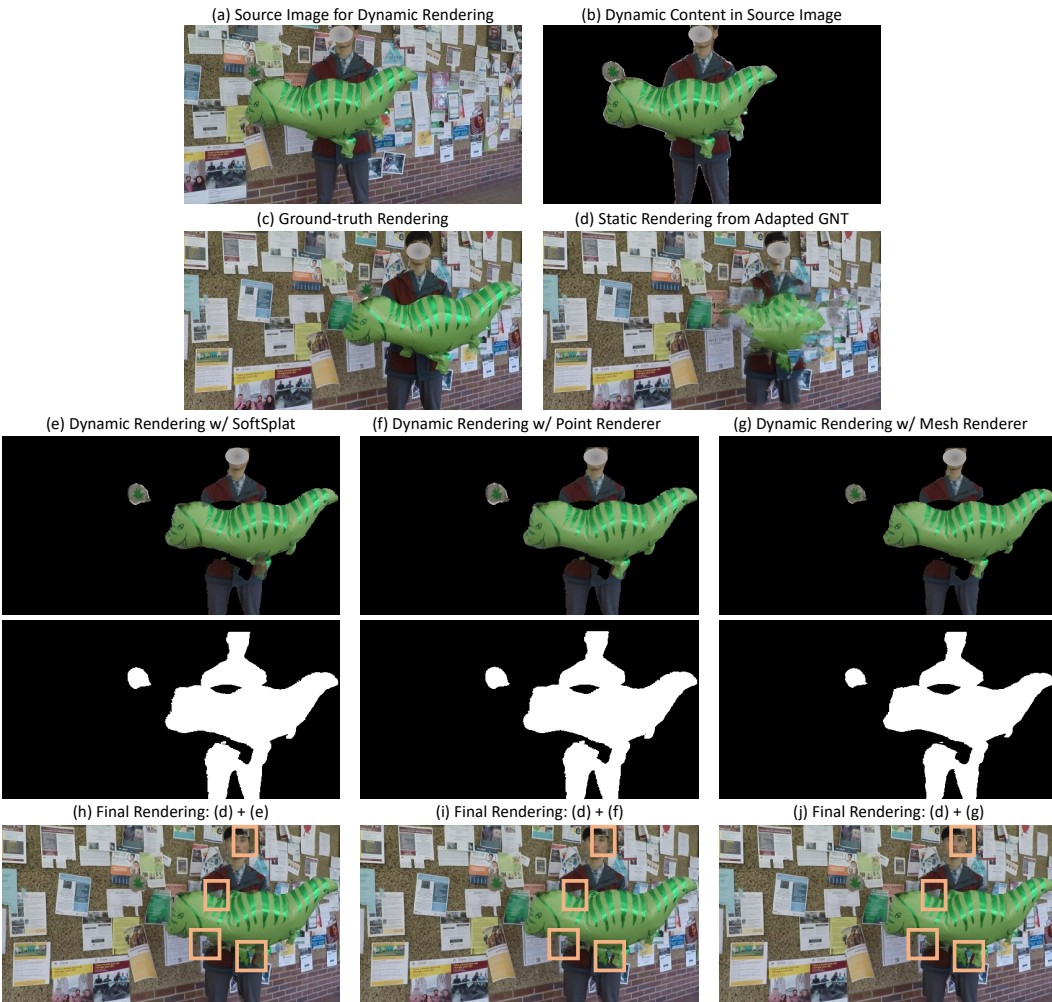

Figure S3: **Artifact analysis.** Due to occlusions in the source image (a), our pseudo-generalized approach is unable to inpaint those highlighted missing areas, resulting in artifacts highlighted in (h) - (j). (e) - (g) display the dynamic content RGB rendering $I_{\text{tgt, dy}}$ and mask $M_{\text{tgt, dy}}$ respectively. When rendering a video, occluded areas change across frames, causing flickering artifacts. Faces are masked to protect privacy.

LPIPS (row 1-3) match the corresponding metrics in v1/2 (row 1-1). Since we reproduce quantitative results by running the official code on rendered images provided by authors, we choose to use our reproduced results for the DVS results in Tab. 1. Namely, we use row 1-4 for DVS results since we evaluate without DynIBaR's masks. Similarly, we use row 2-3 in Tab. S5 for the NSFF results in Tab. 1.

## F    FURTHER ANALYSIS FOR ARTIFACTS

When rendering videos, we observe flickering artifacts. We investigate and find they mainly come from occlusions. To be specific, we display one rendered frame with artifacts in Fig. S3. Fig. S3.(h)-(j) exhibit results from different renderers (see Sec. C.3). This frame is rendered from a camera pose and time step corresponding to the ground-truth in Fig. S3.(c). We also showcase the source image for dynamic rendering in Fig. S3.(a), which is captured at the same time step of Fig. S3.(c). As a result, to obtain the dynamic content rendering, we need to lift the source image's dynamic content (Fig. S3.(b)) with depth, and render it to the target frame. However, comparing Fig. S3.(a) and (c), we can easily detect there exist areas that are unable to be rendered as they are missing

in (a), *e.g.*, the forehead. The dynamic content rendering's mask in Fig. S3.(e) - (g) corroborate our observations. Since occluded areas vary in given monocular videos, such artifacts will change accordingly in renderings, resulting in flickering.

Based on our observations, scene-specific approaches suffer less from flickering artifacts. We believe temporal aggregation techniques can be used to mitigate flickering as we can gather information from more frames for those missing areas. However, as we discussed in Sec. 4.3, simple temporal aggregation with state-of-the-art tracking models does not work well (see Tab. 3 and Tab. S4). More sophisticated designs are required. We leave this to future work.

