# OpenReview forum: "Pseudo-Generalized Dynamic View Synthesis from a Video"
_ICLR.cc/2024/Conference — ICLR 2024 poster_

### Official Review · Reviewer_bWXg · 2023-10-18

**Soundness:** 3 good
**Presentation:** 4 excellent
**Contribution:** 3 good
**Rating:** 8
**Confidence:** 5

**Summary:**

The submission aims to find minimally necessary requirements for a generalized dynamic NeRF method for monocular RGB video input. It finds that running off-the-shelf methods for consistent depth estimation of the RGB video is sufficient. Using the RGB input video and these depth estimates, dynamic novel view synthesis is possible without further per-scene optimization at test time (e.g. for appearance). As additional input annotations, semantic segmentation masks (to identify dynamic objects in the input images) and optical flow are obtained via pretrained, off-the-shelf methods. These inputs can then be combined with a pretrained generalizable static NeRF transformer and novel special handling for dynamic content to render novel views. Experiments on existing benchmark datasets show that the method outperforms or is on par with many prior scene-specific dynamic NeRFs.

**Strengths:**

This is an intriguing problem setting and while the paper does not find a method that is truly fast (due to the consistent depth estimation), it is a carefully executed study with good experiments. It is informative for researchers in the field to see where things currently stand.

The paper is extremely well written. The experiments of the method by itself are very thorough, only comparisons to other methods are a bit lacking (see below).

The appendix is thorough and covers all the questions I had about finer details of the method.

**Weaknesses:**

The most obvious downside of the paper is the result quality, unfortunately. The supplemental videos show rather low-quality results. Given that this is the first method in its problem setting, this is not necessarily a reason for rejection, as long as the experimental evaluation is great. The remaining weaknesses all concern the evluation.

(1) Qualitative video comparisons: I don't understand the results on the Nvidia Dynamic Scenes dataset. Quantitatively, Neural Scene Flow Fields seems to be about on par with the submission. However, qualitatively, the results of NSFF (on their website) are much better. Where does that large discrepancy come from? Qualitative video comparisons to other works would help.

(2) Fast scene-specific dynamic NeRFs: I would like to see a comparison with fast scene-specific dynamic NeRF methods. In terms of utility for novel view synthesis, generalizable NeRFs have two main advantages over scene-specific NeRFs: speed and learned prior knowledge. The latter is not exploited by the submission, as the results show novel view synthesis that sticks closely to the input camera path (instead of revealing hidden areas that scene-specific methods could not handle). Which leaves speed and I'd hence like to a see a comparison with Fang et al. TiNeuVox '22 (code is available), which optimizes a dynamic NeRF in a few minutes, unless there is a reason why such a comparison is unnecessary.

(3) Static generalizable per-frame NVS: Also, given the rather limited quality and temporal instability of the results, I'd like to see a comparison with static single-image generalizable novel view synthesis methods. For example, Sajjadi et al. Scene Representation Transformer '22. MIT-licensed code is here https://github.com/stelzner/srt and the authors say on their project page that this code is reliable.

(4) All qualitative results of the proposed method: Why are the qualitative video results on the DyCheck iPhone dataset not included in the supplement?

== Minor Comments ==

What is "hundreds of GPU hours per video" in the introduction referring to? It sounds as if existing per-scene methods take hundreds of hours per video (incl. appearance optimization). But that's not the case with most monocular dynamic NeRFs, especially the recent fast methods that take minutes rather than hours.

The related work is covered very thoroughly. Only recent diffusion-based approaches for novel view synthesis could additionally be cited, e.g. Watson et al. Novel View Synthesis with Diffusion Models (ICLR '23).

The first citation in A.1 is for transformers in general, not for GNT.

Please add a sentence to the main text that splat-/point-/mesh-based rendering of the point cloud is used to get the dynamic image rendering. The section on dynamic rendering feels incomplete currently, with some context/framing missing.

I would not call the results "high-quality" (e.g. caption of Figure 8). The videos are not high-quality.

**Questions:**

I am confused by what's happening in Table 3. The final method is 3? And 5-1 and 5-2 differ in what way? Does the final method not use Sec. 3.3.2, while 5-1 and 5-2 do?

Other than that, the weaknesses cover the four points I think need to be addressed in a rebuttal. I am open to arguments as to why these experiments might not be necessary.

===

The rebuttal addressed my concerns very well and I am hence updating my score to Accept.

---

> ### Author Response · Authors · 2023-11-13
>
> ## About supplementary videos
>
> Thanks a lot for pointing this out. After double-checking, we found that we indeed uploaded the wrong videos. We corrected this in the updated supplementary material. Videos exhibit much fewer artifacts.
>
> ## About quantitative vs. qualitative results
>
> Note that the videos in the supplementary material are for **spatio-temporal interpolation**, which does not perfectly reflect the quantitative results for **novel-view synthesis** in the main paper.
>
> To be specific, the quantitative results on the multi-view NVIDIA Dynamic Scenes dataset reported in Tab. 1 don’t need temporal interpolation (see the paragraph below Eq. (3)). Essentially, quantitative evaluations are for spatial interpolations. However, when synthesizing the videos in the supplementary material, we conduct both spatial and temporal interpolations to demonstrate the ability of our study-found approach to synthesize frames between observed times. As expected, results are not as good as some methods which use scene-specific optimization. Nonetheless, we think it is valuable to show these results and understand present day capabilities.
>
> To provide illustrations for the quantitative results provided in the main paper, we included videos composed of evaluation frames in the updated supplementary. See folder section “[NVIDIA Dynamic Scenes] Videos from Frames for Quantitative Evaluations” in the updated html.
>
> ## About updated quantitative results
>
> We found an error in computing SSIM results for our approach on NVIDIA Dynamic Scenes data in the original submission. The DyCheck results are not affected.
>
> We corrected it and updated the affected Tab. 1, 3, S1, S2, and S4. After correction, the gap between SSIM of our approach and scene-specific approaches is smaller.
>
> Moreover, we find that DynIBaR’s official evaluations are not fair since they exclude pixels that DynIBaR is unable to render. We added a discussion in the new Sec. E and updated the affected Tab. 1.
>
> ## About videos on NSFF website
>
> Note, the original qualitative results of NSFF and DVS cannot be directly compared here due to the different setup. Originally, NSFF and DVS only selected 24 frames from the full video for training and evaluation. In contrast, we follow DynIBaR and evaluate on the whole video. Arguably, NSFF’s inability to handle long videos is one of the motivations for the development of DynIBaR (see DynIBaR’s Sec. 1 and NSFF results on the DynIBaR website).
>
> ## About fast scene-specific dynamic NeRF
>
> Thanks a lot. Great suggestion. We add quantitative and qualitative results for TiNeuVox on both datasets to the paper and the supplementary html respectively. For qualitative videos, please refer to sections “[NVIDIA Dynamic Scenes] Videos from Frames for Quantitative Evaluations” and “[DyCheck iPhone] Videos from Frames for Quantitative Evaluations” on the updated html.
>
> Specifically, we run TiNeuVox on each video for 40k steps, which is double the steps than the default 20k iterations. This gives TiNeuVox an additional advantage. We evaluate on its checkpoints at both 20k and 40k steps of optimization, which takes 45min and 1.5 hours on a V100.
>
> Note, TiNeuVox can only complete optimization “in a few minutes” on the D-NeRF dataset. According to DyCheck (Gao et al., 2022a), the D-NeRF dataset is arguably one of the easiest datasets for dynamic novel view synthesis as it is “effectively multiview” (see Fig. 3 of DyCheck’s paper). In contrast, the NVIDIA Dynamic Scenes and DyCheck iPhone data are much more challenging.
>
> Meanwhile, we want to clarify that TiNeuVox also requires consistent depth for its regularization loss on background 3D points (https://github.com/hustvl/TiNeuVox/blob/d1f3adb6749420d10ecc074806f06459f189acbd/lib/load_hyper.py#L79-L83). Therefore, our comparison to TiNeuVox is fair.
>
> We observe that TiNeuVox results are inferior to our approach on both datasets.
> We updated Tab. 1, 2 and Fig. 1 to include the updated results.
> Note that there are no big differences between results after 20k and 40k steps, i.e., the optimization converged.
> We hypothesize the inferior results are due to TiNeuVox not disentangling static and dynamic content. Though a joint representation works well on “effective multiview” datasets, e.g., D-NeRF data and Nerfies data, it is more challenging to attain good results with a joint representation on NVIDIA Dynamic Scenes or DyCheck iPhone data.

---

> > ### Author Response · Authors · 2023-11-13
> > **[Continued] Response to Reviewer bWXg**
> >
> > ## About static generalizable per-frame NVS
> >
> > Thanks a lot for the suggestion. We find that the suggested SRT repo (https://github.com/stelzner/srt) only contains pretrained checkpoints on the NMR dataset (a subset of ShapeNet hosted by https://github.com/autonomousvision/differentiable_volumetric_rendering) and the MultiShapeNet (MSN) dataset:
> > - For NMR, as mentioned in the repo (https://github.com/stelzner/srt#known-issues), SRT can only render from fixed 24 camera poses, which is not suitable in our case.
> > - For MSN, which is only trained on ShapeNet objects, a comparison is not suitable due to the large domain gap between ShapeNet and our real world dynamic scenes.
> > Therefore, a comparison to SRT is not meaningful.
> >
> > ## About DyCheck videos
> >
> > Thanks for the suggestion. We add them in the updated supplementary material. See section “[DyCheck iPhone] Videos from Frames for Quantitative Evaluations”.
> >
> > ## About completing main text’s dynamic rendering part
> >
> > Thanks a lot for the suggestion. We clarify in Sec. 3.3.1 of the updated pdf that we use either SoftSplat or point/mesh-based rendering.
> >
> > ## About Fig. 8’s caption
> >
> > Thanks a lot for the suggestion. We changed the word to “reasonable” in the updated pdf.
> >
> > ## About citations
> >
> > Thanks a lot for sharing, we added the missing citation to Sec. 2’s "Generalized static novel-view synthesis” paragraph.
> >
> > We also corrected citations in Sec. A.1. Thanks for pointing it out.
> >
> > ## About 5-1 vs. 5-2 in Tab. 3
> >
> > To clarify, as stated in the title of “Methodology for Studying the Question”, Sec. 3 describes the framework we are using for our study.
> >
> > Our best-performing ablation (row 3 in Tab. 3) does not involve temporal tracking discussed in Sec. 3.3.2, as we find that tracking isn’t as robust as necessary (row 5-1 and 5-2 in Tab. 3).
> >
> > The difference between row 5-1 and 5-2 is the underlying tracking approach. Row 5-1 uses TAPIR while Row 5-2 uses CoTracker. Both are concurrent work and we think it is beneficial to try both to understand the limitations.

---

> > > ### Comment · Reviewer_bWXg · 2023-11-14
> > > **Thank you!**
> > >
> > > I appreciate the rebuttal a lot. It addresses my concerns very well. I am also glad that the corrected result videos look much more temporally stable than before. I am also, at this point, satisfied with the rebuttal to the other reviewers' concerns. Unless a discussion there reveals that I've misunderstood something major, I will increase my rating to Accept.
> > >
> > > The only remaining nitpick I have is that "usually hundreds of hours" really only refers to a few (but impactful) methods like Nerfies or DynIBaR. As far as I've seen, most methods don't take that long because they come from academic labs with fewer resources. It would be nice if this could be qualified in some manner.

---

> > > > ### Author Response · Authors · 2023-11-14
> > > >
> > > > We are thrilled that our rebuttal has successfully addressed your concerns and are encouraged by your decision to raise the score.
> > > >
> > > > We apologize for overlooking this particular point and sincerely appreciate your friendly reminder. We have revised the pdf to rephrase “usually take hundreds of GPU hours” to “could take up to hundreds of GPU hours” in Fig. 1’s caption and the introduction.

---

### Official Review · Reviewer_3K58 · 2023-10-23

**Soundness:** 3 good
**Presentation:** 3 good
**Contribution:** 3 good
**Rating:** 8
**Confidence:** 4

**Summary:**

The paper presents a study on generalized dynamic novel view synthesis from monocular videos, a challenge yet to be addressed in the literature. The authors establish an analysis framework, developing a "pseudo-generalized" technique that doesn't require scene-specific appearance optimization. The study found that geometrically and temporally consistent depth estimates are crucial to achieve this approach. Interestingly, this pseudo-generalized method outperformed some scene-specific techniques.

**Strengths:**

- Originality in addressing the generalized dynamic novel view synthesis from monocular videos.
- Introduction of the pseudo-generalized process without scene-specific appearance optimization.
- A comprehensive set of experiments and detailed ablations to validate the approach.

**Weaknesses:**

- Presentation and clarity can be enhanced.
- A broader range of related works should be included in the comparisons.
- Ambiguity about the role of consistent depth estimates in the final result.
- Experimental validation seems limited to certain datasets, potentially affecting generalizability.

**Questions:**

- Can the authors clarify the specific role and impact of consistent depth estimates in their method?
- How does the proposed method compare to generalized techniques not mentioned in the paper?

---

> ### Author Response · Authors · 2023-11-13
>
> ## About presentation
>
> Thanks a lot for your suggestions. We are more than happy to further polish the paper given specific suggestions.
>
> ## About related works
>
> Thanks a lot for the suggestions. We are happy to add those missing references if specific pointers are given.
>
> ## About role of consistent depth
>
> To clarify, we want to understand whether generalized dynamic novel view synthesis is possible based on state-of-the-art data priors. Based on our study, we find that we are unable to achieve a completely generalized approach. However, with consistent depth, we are able to get rid of scene-specific appearance optimization.
>
> ## About datasets
>
> We evaluate on two challenging datasets: NVIDIA dynamic scenes and DyCheck iPhone. Both are commonly used datasets in the field of dynamic novel view synthesis. Moreover, according to DyCheck (Gao et al., 2022a), these two datasets are much more challenging than others in this field.
>
> ## About other generalized methods
>
> Thanks a lot for the suggestion. We are happy to add more discussions given specific suggestions.

---

> > ### Author Response · Authors · 2023-11-15
> >
> > We hope our rebuttal could answer your questions and could resolve your concerns about our work. Please let us know if there are any further clarifications that we can offer. We’d appreciate a short reply to let us know your thoughts. Thanks a lot for your time.

---

> > > ### Author Response · Authors · 2023-11-20
> > >
> > > Thank you once again for your time and dedication. The discussion period ends in three days. We are more than happy to address any concerns you may have regarding our work. Please let us know if there are any further clarifications that we can offer. Thanks a lot.

---

> > > > ### Author Response · Authors · 2023-11-22
> > > >
> > > > Thank you once again for your valuable time and commitment. We would like to gently note that the discussion period will conclude in around 24 hours. We greatly treasure the chance to respond to any questions or address any concerns you might have about our work. If there are any points that require clarification, please do not hesitate to let us know. Thanks a lot.

---

> > > > > ### Comment · Reviewer_3K58 · 2023-11-22
> > > > > **Thank you!**
> > > > >
> > > > > Thank you so much for discussing these minor concerns. I have no further questions and will stick to my initial positive opinion.

---

### Official Review · Reviewer_BQLw · 2023-10-28

**Soundness:** 2 fair
**Presentation:** 2 fair
**Contribution:** 1 poor
**Rating:** 3
**Confidence:** 5

**Summary:**

The paper aims to solve and study the novel view synthesis problem for a general dynamic scene. It argues that we can have a generalized approach to dynamic novel view synthesis modeling from monocular videos by overcoming the dependence on the scene appearance. The proposed method uses scene depth, optical flow, and dynamic and static content masks, assuming that dynamic motion is linear and spatially consistent. Results on a few datasets are shown to back up the claims made in the paper.

**Strengths:**

* The paper aims at solving a very challenging problem and studies existing bottlenecks.

**Weaknesses:**

## Abstract
- We find a pseudo-generalized … is possible -> We found that … is possible.

## Introduction
- Authors have given explanations justifying the keywords such as generalized, scene-specific optimization, scene consistent depth, etc., used in the paper. Yet, it is rather weak as the approach itself relies on consistent depth estimates of a dynamic scene, which, in fact, is a very open problem and acceptable solutions generally rely on appearance cues and scene flow.

- Furthermore, with years of practice with physical depth sensors—be it iPhone depth sensing modalities or recent LiDAR, it's very hard, if not impossible, to recover consistent depth estimates for outdoor and indoor cluttered scenes. Even for static scenes, it is highly dependent on the subject material type, lighting condition, and other physical phenomena to have acceptable depth from a physical sensor, and here we are dealing with dynamic scenes. This is precisely the reason for methods such as "Stable View Synthesis" CVPR 2021, "Enhanced Stable View Synthesis" CVPR 2023, and Enhancing photorealism enhancement, TPAMI 2022 papers to make use of multi-modal 3D data to train the model. For completion, TPAMI 2022 could also work for dynamic scenes. The paper should emphasize such intrinsic details, detailing the papers mentioned above and the role of 3D data in novel view synthesis.

- Authors should also clarify why MonoNeRF does not qualify the definition of pseudo-generalized approach, given that the paper mentions “it is unclear whether MonoNeRF is entirely generalizable and can remove scene-specific appearance optimization”. It is better to test and present clarity in the rebuttal phase.

## Scene Content Rendering
- “we think it is possible to avoid scene-specific appearance optimization even if the monocular video input contains dynamic objects.” This argument is provided despite the paper relies on Varma et al. 2023 pretrained GNT which greatly benefits from appearance. Please clarify in the rebuttal as it is inconclusive as to how far the proposed methods benefit from Varma et al. 2023 work, given that the current method is aware of the dynamic subject mask. Hence, in my view, the contribution looks very little.

## Using Depth and temporal priors
- The assumptions about linear motion and use of optical flow is mentioned later in the paper. This must be highlighted in the introduction. Also, the assumption about linear dynamics of a scene is not convincing for a paper oriented towards a generalized or pseudo-generalized approach.

## Experiments

- Results are considerably lower in performance. This makes me conclude appearance is indeed an important cue for neural rendering. Of course, it could take more time, yet it helps gain realism. So, I am not sure whether the research presented in the paper is about time optimization or towards photorealistic rendering of dynamic scenes. Please clarify.


- Missing experiments on outdoor dynamic scene dataset such as Cityscapes. Kindly evaluate results on this dataset and compare it with Enhancing photorealism enhancement, TPAMI 2022.

**Questions:**

Kindly refer weakness section.

---

> ### Author Response · Authors · 2023-11-13
>
> ## About generalized approaches
>
> We emphasize that we do not claim our approach is “generalized”. In contrast, we unambiguously highlight our findings: we were unable to achieve a completely generalized approach based on current state-of-the-art data priors (see our abstraction, introduction, and conclusion).
>
> Meanwhile, as correctly mentioned in the review, consistent depth needs scene-specific optimization. Hence we can only achieve a **pseudo-generalized** approach which doesn’t need scene-specific **appearance optimization**.
>
> Importantly, we never state that we do not need appearance cues (note the difference between **appearance cues** and **appearance optimization**). In contrast, we think appearance cues are important and our analysis of GNT adaptation in Sec 3.2.1 is based on exploiting appearance cues.
>
> ## About TPAMI 2022
>
> Thanks a lot for suggesting “Enhancing photorealism enhancement”. As an image-to-image style-transfer work, it requires a synthetic image as input and aims to enhance the photorealism of the input image, which cannot be used for dynamic novel view synthesis.
>
> ## About MonoNeRF
>
> Sorry for the confusion. The word “unclear” reflects our belief that MonoNeRF does not fall in the “generalized” category, as it requires scene-specific optimization on test scenes. To clarify, we copy our explanation from Sec. 2:
> > Notably, MonoNeRF only reports results for a) temporal extrapolation on training scenes; or b) adaptation to new scenes after fine-tuning. Both setups require scene-specific appearance optimization on test scenes.
>
> We point to Fig. 1 and Fig. 5 in the MonoNeRF paper, which verify the need for appearance optimization: without scene-specific appearance optimization, results are not competitive.
>
> ## About the difference to GNT
>
> As an approach proposed for static scenes, GNT cannot handle dynamic novel view synthesis (see row 0 in Tab. S4).
>
> Our adaptation clearly improves the performance of novel view synthesis (see Tab. S4 row 3 vs. row 0 and Fig. S2).
>
> In hindsight, our proposed adaptation for GNT in Sec. 3.2.1 is simple. Yet, we think it is a finding that’s beneficial to the community. As far as we know, there does not exist work that states how to adapt a pretrained generalized view synthesis approach designed for static scenes to dynamic scenes.
>
> ## About mentioning depth and temporal priors in the introduction
>
> We note our use of depth and temporal priors already (see quote below, copied from our submitted introduction). We revised our introduction to make it clearer.
>
> > For the dynamic part, we aggregate dynamic content with the help of two commonly-used data priors, i.e., depth and temporal priors, and study their effects.
>
> ## About linear dynamics
>
> To clarify, we only use the linear motion assumption for dynamics between observed frames. We think this is reasonable since there does not exist any additional information about the dynamics besides the observed frames. Such a technique is commonly used in dynamic novel view synthesis as stated in Sec. 3.3.1:
>
> >  To produce renderings for temporal interpolation, i.e., $t_\text{tgt} \notin \\{ t_1, \dots , t_N \\}$, previous works assume that motion between adjacent observed times is simply linear (Li et al., 2021; 2023).
>
> ## About our contribution
>
> We want to emphasize that our contributions are to study and understand whether a generalized approach for dynamic novel view synthesis is possible nowadays. As a result, the following thought of the reviewer does not fully reflect the goal of our study:
>
> > I am not sure whether the research presented in the paper is about time optimization or towards photorealistic rendering of dynamic scenes.
>
> Without any doubt, with scene-specific appearance optimization we can obtain better results. But this isn’t our goal. As mentioned by reviewer YL5S: “the paper tries to solve a worthwhile problem that would be very impactful to many groups and companies (rerendering a dynamic video without per-video optimization)”. Further note the opinion from reviewer bWXg: this work is a “carefully executed study with good experiments” and “is informative for researchers in the field to see where things currently stand”.
>
> ## About evaluating on Cityscapes
>
> Thanks a lot for the suggestion. As our work focuses on dynamic novel view synthesis, we use the commonly used dataset in the field. Specifically, we evaluate on two datasets: NVIDIA Dynamic Scenes and DyCheck iPhone. According to DyCheck (Gao et al., 2022a), these two datasets are much more challenging than others in this field.

---

> > ### Author Response · Authors · 2023-11-15
> >
> > We hope our rebuttal could answer your questions and could resolve your concerns about our work. Please let us know if there are any further clarifications that we can offer. We’d appreciate a short reply to let us know your thoughts. Thanks a lot for your time.

---

> > > ### Author Response · Authors · 2023-11-20
> > >
> > > Thank you once again for your time and dedication. The discussion period ends in three days. We are more than happy to address any concerns you may have regarding our work. Please let us know if there are any further clarifications that we can offer. Thanks a lot.

---

> > > > ### Author Response · Authors · 2023-11-22
> > > >
> > > > Thank you once again for your valuable time and commitment. We would like to gently note that the discussion period will conclude in around 24 hours. We greatly treasure the chance to respond to any questions or address any concerns you might have about our work. If there are any points that require clarification, please do not hesitate to let us know. Thanks a lot.

---

### Official Review · Reviewer_YL5S · 2023-10-30

**Soundness:** 2 fair
**Presentation:** 4 excellent
**Contribution:** 3 good
**Rating:** 8
**Confidence:** 4

**Summary:**

This paper presents a method to render novel views of a dynamic scene with much less per-scene optimization than competing techniques such as NSFF [Li et al. 2021] and Dynlbar [Li et al. 2023] . The input to the method is a video of a scene and a set of new camera poses over time. The output is a rerendered video.

The method works by computing a mask of the dynamic parts of the scene using existing methods. It also computes depth and optical flow. The dynamic parts of the scene are then modified by turning the dynamic pixels into point clouds and rerendering according to the new camera poses. The static parts of the scene are rerendered using a modified version of the generalizable NeRF transformer [Varma et al. 2023]. The dynamic and static parts of the scene are then combined.

According to the quantitative metrics, the proposed method seems to perform slightly worse than NSFF and better than some other baselines that work on dynamic video inputs. This would be acceptable as the proposed method is much faster than NSFF and Dynlbar. However, qualitatively, according to the supplemental videos, the proposed method seems much worse than all competing methods, with substantial flickering.


Zhengqi Li, Simon Niklaus, Noah Snavely, and Oliver Wang. Neural Scene Flow Fields for SpaceTime View Synthesis of Dynamic Scenes. In CVPR, 2021.

Zhengqi Li, Qianqian Wang, Forrester Cole, Richard Tucker, and Noah Snavely. DynIBaR: Neural Dynamic Image-Based Rendering. In CVPR, 2023.

**Strengths:**

The paper tries to solve a worthwhile problem that would be very impactful to many groups and companies (rerendering a dynamic video without per-video optimization). The paper reads a bit like a systems paper where there are a dozen components (dynamic mask generation, depth estimation, optical flow, static scene rendering, dynamic scene rendering) that contribute to the final solution.

The overall algorithm makes a lot of sense and seems like it should work. There are also a lot of comparisons to other methods and an ablation study.

**Weaknesses:**

Given that Dynlbar exists, the answer to the title’s question seems like a resounding yes? I suggest the authors change the title to not be a general question and to be something specific about how their method works. I know that the authors are not planning on using scene-specific optimizations, which is how they distinguish their work from Dynlbar, but the title does not make this clear.


The supplementary results seem much, much worse than Dynlbar or NSFF. In the presented results, the dynamic portion of the scene flickers in and out of existence. The LPIPS metrics reported in the paper (Figure 1, Table 1) are only slightly worse than NSFF, but the actual results seem much worse. Perhaps this is because LPIPS doesn’t capture any notion of temporal consistency between the frames? The presented results are very inconsistent while those in NSFF and Dynlbar are not that inconsistent.

I am not sure if the metrics evaluated make sense since they don’t take into account temporal consistency.

There is a minor missing citation. Consider discussing Figure 5 of https://arxiv.org/pdf/1909.05483.pdf [Niklaus et al. 2019] when presenting the statistical outlier removal technique (Fig.~S1). Niklaus et al. 2019 solve a similar problem where inaccurate depth estimates at object boundaries cause a similar problem to the one presented in Figure S1.

My relatively negative rating is based on the seemingly low quality results presented in the supplemental. It seems like the proposed technique does not work that well? The overall algorithm makes sense to me, so I am very surprised at how low quality the results are.

**Questions:**

The video results presented in the supplemental seem much worse than both NSFF and Dynlbar. Specifically, there is a lot of temporal flickering in the dynamic parts of the scene. Do the authors know why this is? I would be very interested in seeing the dynamic mask, the optical flow and the depth estimates to better understand why there is so much flickering. I wonder if the authors uploaded the wrong set of results?

I would be interested in seeing video results on the DyCheck dataset.

How does the proposed method handle cases where the new camera poses peer behind an object to a location that was not seen in any of the input frames? There’s no explicit inpainting step, but my guess is that GNT will not do a good job in these locations? You can kind of see this issue in the presented videos at the edge of the frames, where there are parts of the scene not seen in any of the input views.

---

> ### Author Response · Authors · 2023-11-13
>
> ## About title
>
> Thanks a lot for the suggestion. We plan to change the title to “Pseudo-generalized dynamic novel view synthesis without scene-specific appearance optimization”.
>
> ## About supplementary videos
>
> Thanks a lot for pointing this out. After double-checking, we found that we indeed uploaded the wrong videos. We corrected this in the updated supplementary material. Videos exhibit much fewer artifacts.
>
> ## About quantitative vs. qualitative results
>
> Note that the videos in the supplementary material are for **spatio-temporal interpolation**, which does not perfectly reflect the quantitative results for **novel-view synthesis** in the main paper.
>
> To be specific, the quantitative results on the multi-view NVIDIA Dynamic Scenes dataset reported in Tab. 1 don’t need temporal interpolation (see the paragraph below Eq. (3)). Essentially, quantitative evaluations are for spatial interpolations. However, when synthesizing the videos in the supplementary material, we conduct both spatial and temporal interpolations to demonstrate the ability of our study-found approach to synthesize frames between observed times. As expected, results are not as good as some methods which use scene-specific optimization. Nonetheless, we think it is valuable to show these results and understand present day capabilities.
>
> To provide illustrations for the quantitative results provided in the main paper, we included videos composed of evaluation frames in the updated supplementary. See folder section “[NVIDIA Dynamic Scenes] Videos from Frames for Quantitative Evaluations” in the updated html.
>
> Note, the original qualitative results of NSFF and DVS cannot be directly compared here due to the different setup. Originally, NSFF and DVS only selected 24 frames from the full video for training and evaluation. In contrast, we follow DynIBaR and evaluate on the whole video. Arguably, NSFF’s inability to handle long videos is one of the motivations for the development of DynIBaR (see DynIBaR’s Sec. 1 and NSFF results on the DynIBaR website).
>
> ## About updated quantitative results
>
> We found an error in computing SSIM results for our approach on NVIDIA Dynamic Scenes data in the original submission. The DyCheck results are not affected.
>
> We corrected it and updated the affected Tab. 1, 3, S1, S2, and S4. After correction, the gap between SSIM of our approach and scene-specific approaches is smaller.
>
> Moreover, we find that DynIBaR’s official evaluations are not fair since they exclude pixels that DynIBaR is unable to render. We added a discussion in the new Sec. E and updated the affected Tab. 1.
>
>
> ## About evaluation metrics for temporal consistency
>
> We think better LPIPS indicates a better alignment with the ground truth, which implicitly reflects better consistency. Meanwhile, we agree that LPIPS cannot specifically assess consistency. As far as we know, there are no commonly-accepted metrics to assess the consistency of dynamic novel view synthesis. Any suggestions are very welcome.
>
> ## About the consistency of DynIBaR
>
> We emphasize that it is expected that our study-found approach performs worse than DynIBaR. As stated in the paper (Sec. 4.2), DynIBaR performs appearance optimization using over 300 GPU hours per video while we don’t use any appearance optimization.
>
> ## About understanding flickering
>
> We added a new Sec. F in the updated pdf to analyze and understand the flickering. We find flickering essentially comes from occlusions.
>
> ## About video results on DyCheck
>
> We added videos composed of evaluation frames in the updated supplementary material. See section “[DyCheck iPhone] Videos from Frames for Quantitative Evaluations” in the updated html.
>
> ## About view synthesis from cameras behind objects
>
> You are right that our approach cannot handle such challenging scenarios. Context-aware generative priors are required to inpaint those missing regions as we state in Sec. 4.4.
>
> ## About missing citations
>
> We added a discussion in Sec. C2 of the updated pdf.

---

> > ### Author Response · Authors · 2023-11-15
> >
> > We hope our rebuttal could answer your questions and could resolve your concerns about our work. Please let us know if there are any further clarifications that we can offer. We’d appreciate a short reply to let us know your thoughts. Thanks a lot for your time.

---

> > > ### Author Response · Authors · 2023-11-20
> > >
> > > Thank you once again for your time and dedication. The discussion period ends in three days. We are more than happy to address any concerns you may have regarding our work. Please let us know if there are any further clarifications that we can offer. Thanks a lot.

---

> > > > ### Author Response · Authors · 2023-11-22
> > > >
> > > > Thank you once again for your valuable time and commitment. We would like to gently note that the discussion period will conclude in around 24 hours. We greatly treasure the chance to respond to any questions or address any concerns you might have about our work. If there are any points that require clarification, please do not hesitate to let us know. Thanks a lot.

---

> > > > > ### Comment · Reviewer_YL5S · 2023-11-22
> > > > >
> > > > > Thank you for the rebuttal. It has addressed my concerns about the paper and I see that the concerns brought up by other reviewers is also addressed.

---

### Author Response · Authors · 2023-11-13
**Thanks to All Reviewers**

We thank all reviewers for their time and effort in reviewing the paper.

We are encouraged that reviewers acknowledge that we study “a worthwhile problem that would be very impactful to many groups and companies”  (YL5S) and that we “aim at solving a very challenging problem and study existing bottlenecks” (BQLw).

We are happy that reviewers praise the “originality in addressing the generalized dynamic novel view synthesis from monocular videos” (3K58) and appreciate that “this is the first method in its problem setting” (bWXg).

We are delighted to see reviewers find this work to be “a carefully executed study with good experiments. It is informative for researchers in the field to see where things currently stand” (bWXg).

We address reviewer questions below.

---

### Meta-Review · Area_Chair_6jE2 · 2023-12-08

**Metareview:**

This is an interesting paper which explores the limits of dynamic novel-view synthesis with off-the-shelf systems and w/o video-specific finetuning. While there is an obvious limitation of the work in terms of the result quality compared to prior work that performs video-specific neural optimization (e.g. DynIBaR), the proposed approach is far simpler and computationally efficient.

While the paper received mixed reviews (3xAccept, 1xReject), the AC felt that the concerns by the reviewer recommending rejection were largely addressed in the author response and the AC would concur with the 3 reviewers recommending accept. In particular, while the presented system might not be the most useful one (except as a fast baseline), this discussion in this work maybe helpful for the community and spur the search for non-optimization based solutions.

Finally, the AC would strongly encourage the authors to follow through on the title change they agreed to in the reviewer comments in presenting the work not as a question, but as a simple approach for pseudo-generalizable view synthesis.

**Justification For Why Not Higher Score:**

While the paper is an interesting and can serve as a strong baseline as well as raise interesting discussion points in the community, the quality of results is rather limited and the overall system may not be the most impactful one.

**Justification For Why Not Lower Score:**

Overall, the presented approach of optimization-free dynamic view synthesis is an interesting and efficient one. Given that current approaches predominantly focus on per-video optimization, this paper raises a timely point that this need not be the case and proposes a (slightly worse) efficient baseline. The AC feels this can be a valuable contribution and a discussion point in a rapidly growing area.

---

### Decision · Program_Chairs · 2024-01-16

Accept (poster)